# A is for Absorption: Studying Feature Splitting and Absorption in Sparse Autoencoders

**David Chanin**[*,1,2]     **James Wilken-Smith**[*,1]     **Tomáš Dulka**[*,1]     **Hardik Bhatnagar**[*,1,3]
**Satvik Golechha**[4]     **Joseph Bloom**[1,5]

[1]LASR Labs     [2]University College London     [3] Tübingen AI Center, University of Tübingen
[4]MATS     [5]Decode Research

## Abstract

Sparse Autoencoders (SAEs) aim to decompose the activation space of large language models (LLMs) into human-interpretable latent directions or features. As we increase the number of features in the SAE, hierarchical features tend to split into finer features ("math" may split into "algebra", "geometry", etc.), a phenomenon referred to as feature splitting. However, we show that sparse decomposition and splitting of hierarchical features is not robust. Specifically, we show that seemingly monosemantic features fail to fire where they should, and instead get "absorbed" into their children features. We coin this phenomenon *feature absorption*, and show that it is caused by optimizing for sparsity in SAEs whenever the underlying features form a hierarchy. We introduce a metric to detect absorption in SAEs, and validate our findings empirically on hundreds of LLM SAEs. Our investigation suggests that varying SAE sizes or sparsity is insufficient to solve this issue. We discuss the implications of feature absorption in SAEs and some potential approaches to solve the fundamental theoretical issues before SAEs can be used for interpreting LLMs robustly and at scale.

## 1   Introduction

Large Language Models (LLMs) have achieved remarkable performance across a wide range of tasks, yet our understanding of their internal mechanisms lags behind their capabilities. This gap between performance and interpretability raises concerns about the "black box" nature of these models [30]. The field of mechanistic interpretability aims to address this issue by reverse-engineering the internal algorithms of neural networks and performing causal analysis on them [24].

Recent work theorizes that models represent concepts as linear directions in a high-dimensional space, known as the Linear Representation Hypothesis (LRH) [26, 8]. The model is able to represent far more concepts than it has neurons in its hidden space by allowing these concept directions to overlap slighly, known as superposition [8]. Superposition makes it challenging to directly interpret neurons in an LLM, and requires different techniques to extract interpretable feature directions.

As long as the active features in a given LLM activation are sparse and the underlying features follow the LRH, Sparse Autoencoders (SAEs) should be able to recover the true LLM features despite supersition using sparse dictionary learning [27]. Indeed, SAEs have shown potential in decomposing the dense, polysemantic activations of LLMs into more "interpretable" latent features [5, 2].

However, we show that even if all underlying features are linear and sparsely activating, an SAE will still fail to recover the true underlying features if the features form a hierarchy. Instead, an SAE will

---

[*]These authors contributed equally to this work.

39th Conference on Neural Information Processing Systems (NeurIPS 2025).

| **Ideal interpretable solution** | | | **Uninterpretable absorption solution** | | |
|---|---|---|---|---|---|
| | SAE Encoder | SAE Decoder | | SAE Encoder | SAE Decoder |
| Latent 1 | "starts with S" | "starts with S" | Latent 1 | ¬"short" ∧ "starts S" | "starts with S" |
| Latent 2 | "short" | "short" | Latent 2 | "short" | "short" + "starts S" |

| The ideal interpretable solution requires firing **2** latents to represent "short". | Absorption only requires firing **1** latent to represent "short", and is unfortunately what the SAE learns. |
|---|---|

Figure 1: In feature absorption, seemingly monosemantic latents fail to fire in cases where they apparently should. Here, we see an SAE can represent the word "short" and the concept "starts with S" more sparsely by absorbing the "starts with S" direction into the "short" latent, and then not firing the "starts with S" latent on the word "short", despite "short" starting with "S". Logical notation is used to describe the SAE encoder to emphasize its role as a classifier.

learn gerrymandered latents that fail to fire on seemingly arbitrary cases where the latent should fire according to the mainline interpretation of the latent. We refer to this failure as feature absorption.

Feature absorption is demonstrated in Figure 1, where the feature "short" always fires alongside a feature representing "starts with S". Instead of learning an interpretable latent representing "starts with S", the SAE can increase sparsity by instead disabling the "starts with S" latent when "short" is active while still getting perfect reconstruction.

We present the following contributions: (1) we identify a problematic variant of feature-splitting we call "feature absorption", where an SAE latent appears to track a human-interpretable concept, but fails to activate on seemingly arbitrary tokens. Instead, more specific latents activate and contribute a component of feature direction, "absorbing" the feature. (2) We demonstrate that feature absorption is caused by hierarchical features. (3) We develop a metric to detect feature absorption in LLM SAEs. And (4) we validate that feature absorption occurs in every LLM SAE we tested, including hundreds of open-source SAEs.

Feature absorption poses an obstacle to the practical application of SAEs since it suggests SAE latents may be inherently unreliable classifiers. This is particularly important for applications where we need confidence that latents are fully tracking behaviors, such as bias or deceptive behavior. Furthermore, techniques which seek to describe circuits in terms of a sparse combination of latents will also be more difficult in the presence of feature absorption [21].

An online explorer for our results can be found at `https://feature-absorption.streamlit.app`. Code is available at `https://github.com/lasr-spelling/sae-spelling`.

## 2   Background

**Hierarchical features.**   We say features $f_1$ and $f_2$ form a hierarchy with $f_1$ as the parent and $f_2$ as the child if $f_2 \implies f_1$, meaning every time $f_2$ fires $f_1$ must also fire.

**Linear probing.**   A linear probe is a simple linear classifier trained on the hidden activations of a neural network, typically using logistic regression (LR) [1].

**K-sparse probing.**   A k-sparse probe [12] is a linear probe trained on a sparse subset of $k$ neurons or SAE latents. Training a k-sparse probe first requires selecting the $k$ best neurons or SAE latents that in-aggregate act as a good classifier, and then training a standard linear probe on just those $k$ neurons or latents.

Gurnee et al. [12] proposed several methods of estimating the best $k$ neurons or latents to pick, one of which involves first training a LR probe with a L1 loss term, and selecting the $k$ largest elements by probe weight. When we refer to k-sparse probing in this work, we use this method of selecting $k$ latents.

**Sparse autoencoders.**   An SAE consists of an encoder, $W_{enc}$, a decoder, $W_{dec}$, and corresponding biases $b_{enc}$ and $b_{dec}$. The SAE has a nonlinearity, $\sigma$, typically a ReLU (or variant such as JumpReLU [29]). Given input activation, $a$, the SAE computes a hidden representation, $f$, and reconstruction, $\hat{a}$:

$$f = \sigma(W_{enc}a + b_{enc}) \tag{1}$$
$$\hat{a} = W_{dec}f + b_{dec} \tag{2}$$

SAEs attempt to reconstruct input activations by projecting into an overcomplete basis using a sparsity-inducing loss term (typically $L1$ loss), or a certain number of non-zero latents ($L0$) on the hidden activations.

**SAE feature ablation.** In an ablation study we examine the downstream causal effect of an SAE latent by computing how patching its activation to 0 changes a downstream metric (e.g. logit difference). A negative ablation effect means intervening on the SAE latent would lower the metric. We follow the work of Marks et al. [21] and provide the algorithm in Appendix A.4.

## 3 Toy models of feature absorption

Absorption is caused by the SAE sparsity penalty in the presence of hierarchical features. When two features form a hierarchy, for instance "starts with S" and "short", the SAE can merge the "starts with S" feature direction into a latent tracking "short" and then not fire the main "starts with S" latent. This means firing one latent instead of two, increasing sparsity while retaining perfect reconstruction. We demonstrate that hierarchical feature co-occurrence causes absorption in a simple toy setting.

Our initial setup consists of 4 true features, each randomly initialized into orthogonal directions with a 50 dimensional representation vector and unit norm. Each feature fires with magnitude 1.0. Feature $f_0$ fires with probability 0.25, and features $f_1$, $f_2$, and $f_3$ fire with probability 0.05. Each SAE training input is created by sampling from these true features and summing the directions of each firing feature. We train a SAE with 4 latents to match the 4 true features using SAELens [15]. The SAE uses L1 loss with L1 coefficient 3e-5, and learning rate 3e-4. We train on 100M activations.

**Independently firing features** When the true features fire independently, we find that the SAE is able to perfectly recover these features as shown in Figure 2a. The SAE learns one latent per true feature. The decoder representations perfectly match the true feature representations, and the encoder learns to perfectly segment out each feature from the other features.

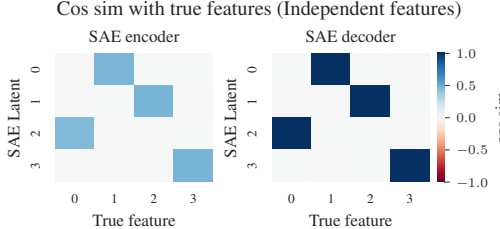
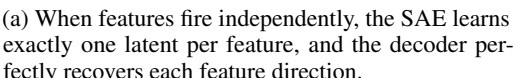
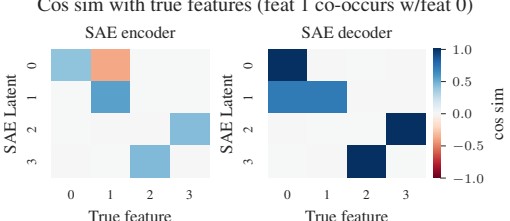

(a) When features fire independently, the SAE learns exactly one latent per feature, and the decoder perfectly recovers each feature direction.

(b) When features 0 and 1 co-occur (feature 1 only fires if feature 0 fires), we see absorption in the SAE encoder and decoder latents 0 and 1.

Figure 2: Comparison of independent features (left) vs. co-occurring features with absorption (right)

**Hierarchical features cause absorption** Next, we modify the firing pattern of feature 1 so it fires only if feature 0 also fires. We keep the overall firing rate of feature 1 the same as before, firing in 5% of activations. Features 2 and 3 remain independent.

Figure 2b shows the encoder and decoder cosine similarities with the true features in the hierarchical co-occurrence setup. Here, we see a clear example of feature absorption. Latent 0

Figure 3: Interpretation of learned SAE latents with co-occurrence between feature 0 and feature 1 (feature 1 only fires if feature 0 fires).

|  | SAE ENCODER | SAE DECODER |
|---|---|---|
| LATENT 0 | ¬ feat 1 ∧ feat 0 | feat 0 |
| LATENT 1 | feat 1 | feat 0 + feat 1 |
| LATENT 2 | feat 3 | feat 3 |
| LATENT 3 | feat 2 | feat 2 |

has learned a perfect representation of feature 0, but the encoder has a hole in its recall. Latent 0 fires if feature 0 is active but not feature 1. This is exactly the sort of gerrymandered feature firing pattern we will see later in real SAEs in Section 5.2 - the encoder has learned to stop the latent firing on specific cases where it looks like it should be firing. In addition, we see that latent 1, which tracks feature 1, has absorbed the feature 0 direction. This results in latent 1 representing a combination of feature 0 and feature 1. We see that the independently firing features 2 and 3 are untouched - the SAE still learns perfect representations of these features. These results are summarized in Table 3. We explore absorption in more toy settings in Appendix A.3.

**Proof: hierarchical features cause absorption**     We further provide an analytical proof that in the hierarchical setup described above, feature absorption decreases SAE loss in Appendix A.2.

## 4    Experimental setup

Our experiments on LLM SAEs focus on predicting the first-letter of a single token containing characters from the English alphabet (a-z, A-Z) and an optional leading space. We use in-context learning (ICL) prompts to elicit knowledge from the model, using templates of the form:

```
{token} has the first letter:  {capitalized_first_letter}
```

An example of an ICL prompt consisting of 2 in-context examples is shown below. The model should output the _D token:

```
 tartan has the first letter: T
 mirth has the first letter: M
 dog has the first letter:
```

In the above prompt, we extract residual stream activations at the _dog token index. These activations are used both for LR probe training and for applying SAEs. We use a train/test split of 80% / 20%, and evaluate only on the test set of the probes, including when running experiments on SAEs. When applying SAEs, we include the SAE error term [21] to avoid changing model output.

To determine the causal effect of SAE latents on the first-letter identification task we conduct ablation studies. We use a metric consisting of the logit of the correct letter minus the mean logit of all incorrect letters. This measures the propensity of the model to choose the correct starting letter as opposed to other letters. Formally, our metric $m$ is defined below, where $g$ refers to the final token logits, $L$ is the set of uppercase letters, and $y$ is the uppercase letter that is the correct starting letter:

$$m = g[y] - \frac{1}{|L| - 1} \sum_{l \in \{L \setminus y\}} g[l]$$

We discuss this metric and alternative formulations further in Appendix A.10.

To determine how well multiple latents perform as a classifier when used together, we use k-sparse probing, increasing the value of $k$ from 1 to 15. We train a LR probe using a L1 loss term with coefficient 0.01, and select the top $k$ latents by magnitude.

We use the base Gemma-2-2B model for most of our studies, along with the full set of Gemma Scope residual stream SAEs of width 16k and 65k released by Deepmind [19]. We also evaluate absorption on our own SAEs trained on Qwen2 0.5B [32] and Llama 3.2 1B [6].

## 5    Results

Our results are divided into four sections. First, we compare the performance of linear probes with SAE latents on recovering first-character information from model activations, showing that despite appearing to track first letter features, a wide variety of precision / recall is achieved. Second, we motivate our definition of feature absorption with a case-study, emphasizing how an absorbing latent can unexpectedly causally mediate first letter information whilst the first-letter latent (unexpectedly) fails to fire. Next, we attempt to quantify feature splitting and feature absorption, showing that tuning of hyper-parameters may partially assist but not fully alleviate feature absorption.

## 5.1 Do SAEs learn latents that track first letter information?

We compare the performance of LR probes with the performance of the SAE latent whose encoder direction has highest cosine similarity with the probe, resulting in 26 "first-letter" latents. We observed that for each probe, there was clearly one or at most a couple of outlier SAE latents with high probe cosine similarity. Full plots of cosine similarity vs letter are shown in Appendix A.8.

We also tried using k=1 sparse probing [12] to identify SAE latents, and found this gives similar results. Further comparison of using k=1 sparse probing vs encoder cosine similarity to identify latents is explored in Appendix A.7.

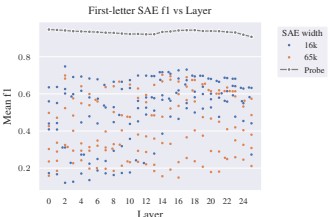

(a) Mean F1 score using top SAE encoder latent by layer for Gemma Scope SAEs and LR probe.

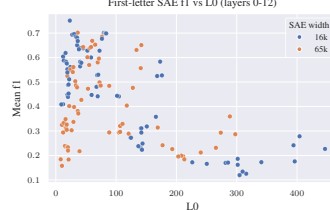

(b) Mean F1 score vs L0 for Gemma Scope SAEs layers 0-12. SAEs learn cleanest "first-letter" features with L0 near 25-50.

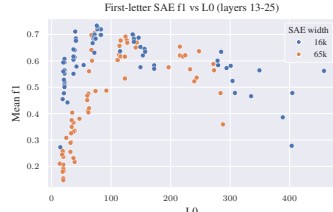

(c) Mean F1 score vs L0 for Gemma Scope SAEs layers 13-25. SAEs learn cleanest "first-letter" features with L0 near 50-100.

Figure 4: Comparison of F1 scores for first-letter classification tasks

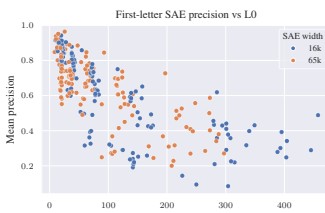

(a) Mean precision using top SAE encoder latent vs L0 for all Gemma Scope SAEs.

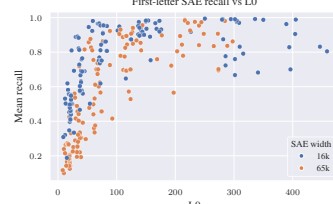

(b) Mean recall using top SAE encoder latent vs L0 for all Gemma Scope SAEs.

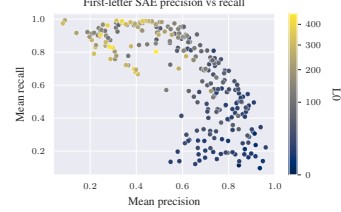

(c) Mean precision vs recall using top SAE encoder latent vs L0 for all Gemma Scope SAEs.

Figure 5: Precision and recall vs L0 for first-letter classification tasks

We observe wide variance in the performance of Gemma Scope SAEs at the first-letter identification task, but no SAE matches LR probe performance. We show the mean F1 score by layer as well as the F1 score of the LR probe in Figure 4a. We further investigate the F1 score of these SAE encoder latents as a function of L0 and SAE width in Figures 4b and 4c.

Whether or not an SAE learns a clear "first-letter" latent for each letter is highly dependent on L0, with low L0 SAEs tending to learn high-precision low-recall latents, and high L0 SAEs learning low-precision high-recall latents (Figure 5). We caution drawing conclusions about an "optimal" L0 from these plots, as we find further variance when broken-down by letter, shown in Appendix A.8.

## 5.2 Why do SAE latents underperform?

The Gemma Scope layer 3, 16k width, 59 L0 SAE has a latent, 6510, which appears to act as a classifier for "starts with S", achieving an F1 of 0.81. However, this latent fails to activate on some tokens the probe can classify, and which the model can spell, such as the token _short.

Figure 6a shows a sample prompt containing a series of tokens that start with "S", and the activations of top SAE latents by ablation score for these tokens. The main "starts with S" latent, 6510, activates on all these tokens except _short. This SAE also has a token-aligned latent, 1085, which activates on variants of the word "short" (" short", "SHORT", etc...). The Neuronpedia dashboard [20] for latent 1085 is shown in Appendix A.15. For the token _short, the main "starts with S" latent does not activate but the "short" latent activates instead.

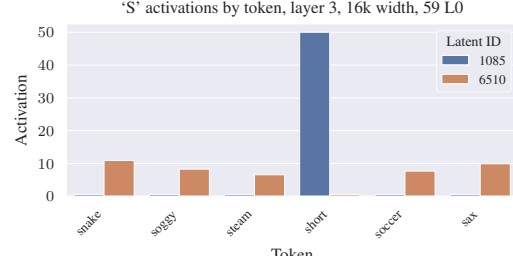

(a) Layer 3, L0=59 SAE latent activations for tokens that start with "S". The core "starts with S" latent, 6510, fails to activate on the token `_short`. The "short"-token aligned latent 1085 activates instead.

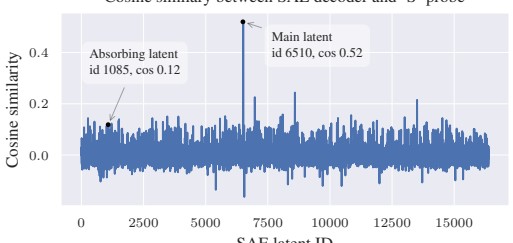

(b) Cosine similarity between layer 3, L0=59 SAE decoder and the "starts with S" probe. The main "Starts with S" latent, 6510, is clearly visible and highly probe-aligned.

Figure 6: Comparison of SAE latent activations and cosine similarity for tokens starting with "S"

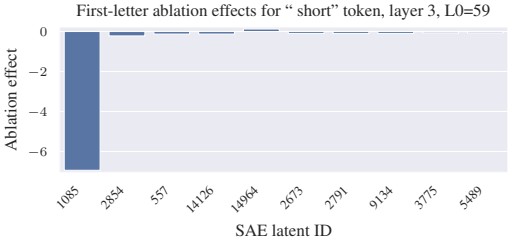

(a) Ablation effect for `_short` token, indicating that latent 1085, is responsible for the "starts with S" concept for the `_short` token. The main "starts with S" latent, 6510, does not activate on the `_short` token.

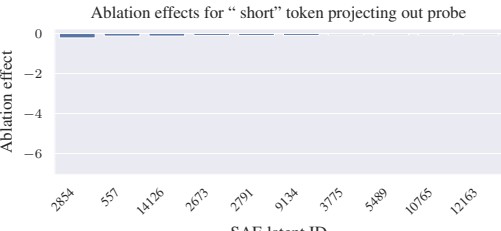

(b) Ablation effect for `_short` token after removing the probe direction from latent 1085 via projection. Latent 1085 no longer appears in the plot, indicating the strong ablation effect in Figure 7a is due to its component along the probe direction.

Figure 7: Ablation effects on `_short` token before and after projecting out the probe direction

Latent 1085 has a cosine similarity with the "starts with S" probe of 0.12, indicating it contains a component of the "starts with S" direction, although much smaller than the main "starts with S" latent. Cosine similarity of the SAE decoder with the "starts with S" LR probe is shown in Figure 6b. Interestingly, despite latent 1085 having only about 1/5 the cosine similarity with the probe as the main latent 6510, we see it activates with about 5 times the magnitude of latent 6510 on the `_short` token, thus contributing a similar amount of the "starts with S" probe direction to the residual stream.

We study the ablation effect of each SAE latent on the `_short` token, shown in Figure 7a, and see that latent 1085 has a dramatically larger ablation effect compared with all other SAE latents. This suggests latent 1085 is causally responsible for the model knowing that `_short` starts with S.

Is it possible that the probe projection is not the causally important component of latent 1085? We conduct another ablation effect experiment, except now we remove the probe direction from latent 1085 via projection before ablation. The results of this experiment are shown in Figure 7b. After removing the probe component from latent 1085, it no longer has a significant ablation effect. Thus we know the probe projection of latent 1085 is responsible for model behavior.

These experiments show the "starts with S" feature has been "absorbed" by the token-aligned latent 1085, likely along with other semantic concepts related to the word "short". After observing that the main "starts with S" latent 6510 activates on most tokens that begin with "S", it may be tempting to conclude this latent tracks the interpretable feature of beginning with the letter "S". However, this latent quietly fails to activate on the `_short` token, leading us to a false sense of understanding.

Here we clearly see feature absorption. The seemingly interpretable SAE latent 6510 fails to activate on arbitrary positive examples, and instead the feature is "absorbed" into more specific latents.

Feature absorption is likely a logical consequence of SAE sparsity loss. If a dense and sparse feature co-occur, absorbing the dense feature into a latent tracking the sparse feature will increase sparsity.

Table 1: Sample max activating examples for latents 7112 and 7657 for Gemma Scope 16k, layer 0, 105 L0 from Neuronpedia. The token where the SAE latent activates is highlighted in yellow. Latent 7112 appears to be a lowercase "L" starting-letter latent, and latent 7657 appears to be a corresponding uppercase "L" latent.

| LATENT 7112 | LATENT 7657 |
| --- | --- |
| žda se napla**ća**uje naknada | **LC**, an aluminum boat |
| . E. Sø**li**, 20 | as **LIFT** and **LF**-Net. Once |
| a></**li**></ul | latter's sister **Louise**, who in |

## 5.3 Measuring feature splitting and feature absorption

**Feature splitting**   A key phenomenon identified from previous studies of SAEs is feature-splitting [2], where a feature represented in a single latent in a smaller SAE can split into two or more latents in a larger SAE. During our experiments, we found strong evidence of feature-splitting in the Gemma Scope SAEs.

For instance, in the layer 0, 16k width, 105 L0 SAE, we find two encoder latents (id:7112 and id:7657 [1]) which align with the "L" starting letter probe. Inspecting max activating examples, we see latent 7112 activates on tokens starting with lowercase "l", while 7657 activates on tokens starting with uppercase "L". Some activating examples for these latents are shown in Table 1.

Feature splitting like this is not necessarily problematic for interpretability efforts since the split features are still easily identifiable, and depending on the context it may be more useful to have either a single "starts with L" latent or a pair of "starts with uppercase / lowercase L" latents.

We measure feature splitting using k-sparse probing [12] on SAE activations. If increasing the k-sparse probe from $k$ to $k+1$ causes a significant increase in probe F1 score, then the additional SAE latent provides a meaningful signal, and the combination of these $k+1$ latents is likely a feature split. In the example of the uppercase "L" and lowercase "l" split, a k-sparse probe with $k=2$ trained on both these latents should predict "starts with letter L" much better than either latent on its own. Figure 8a shows F1 vs K for letters "L" and "N". The "L" k-sparse probe shows a significant jump in F1 score moving from k=1 to k=2 corresponding to feature splitting, while the F1 score for the "N" k-sparse probe is relatively constant.

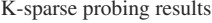

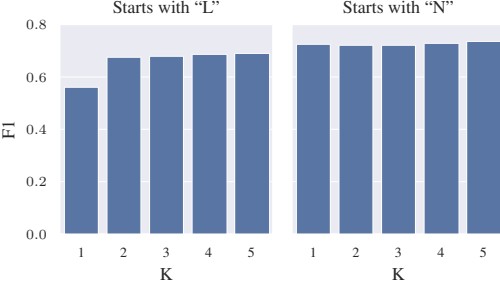

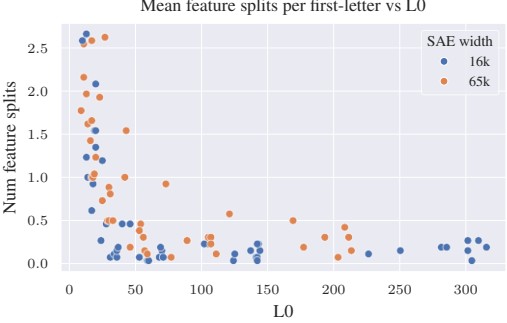

(a) K-sparse probing results for letters "L" and "N", layer 0, 16k width, 105 L0. "L" shows a significant improvement in F1 between k=1 and k=2 corresponding to feature splitting.

(b) Mean number of feature splits per letter on the first-letter spelling task by L0. Feature splitting occurs more frequently with higher sparsity.

Figure 8: Feature splitting analysis in sparse autoencoders

We detect feature splitting by measuring whether increasing $k$ by one causes a jump in F1 score by more than a threshold, $\tau$. We use $\tau = 0.03$ as a reasonable choice after inspecting situations like in Figure 8a, where feature splitting corresponds to an F1 score jump between 0.05 - 0.1. Figure 8b shows feature splitting vs L0 for all 16k and 65k width Gemma Scope SAEs.

---

[1] `https://www.neuronpedia.org/list/cm0h1n2mt00019jdk274owq9e`

The single latent or a set of traditional feature split latents that seem to act as a classifier for a human-interpretable feature like "starts with S" fail to fire in a seemingly arbitrary number of cases. What fires instead are often approximately token-aligned latents with small but positive alignment with the LR probe. We say these latents are absorbing the feature.

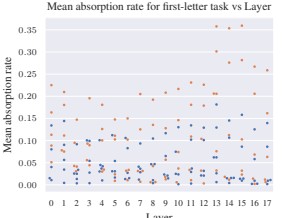 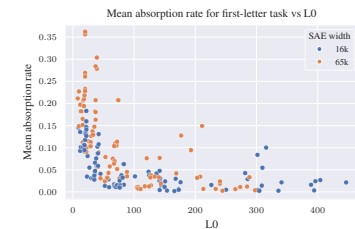 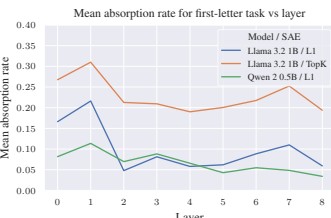

(a) Mean feature absorption rate vs layer on the first-letter task, Gemma Scope 16k and 65k SAEs. We do not see an obvious pattern in absorption rates by layer.

(b) Mean feature absorption rate vs L0 on first-letter task, Gemma Scope 16k and 65k SAEs. Wider and more sparse SAEs demonstrate higher rates of absorption.

(c) Mean feature absorption rate vs layer on the first-letter task on Llama 3.2 1B and Qwen 2 0.5B models, L1 loss and TopK SAE architectures, layers 0-8.

Figure 9: Feature absorption rates

We quantify the extent to which feature absorption occurs with the metric **feature absorption rate**. We first find $k$ feature splits for a first-letter feature using a k-sparse probe. We then find false-negative tokens that all $k$ feature-split SAE latents fail to activate on, but which the LR probe correctly classifies, and run an integrated-gradients ablation experiment on those tokens. The ablation effect finds the most causally important SAE latents for the spelling of that token. If the SAE latent receiving the largest negative magnitude ablation effect has a cosine similarity with the LR probe above $0.025$, and is at least $1.0$ larger than the latent with the second highest ablation effect, we say that feature absorption has occurred. These thresholds were chosen from manual inspection of the data to best distinguish the absorption phenomenon. We then calculate feature absorption rate as:

$$\texttt{absorption\_rate} = \frac{\texttt{num\_absorptions}}{\texttt{lr\_probe\_true\_positives}}$$

If there are more than 200 false negatives per letter, we randomly pick 200 samples to estimate the number of absorptions. We see absorption rate increases with higher sparsity and higher SAE width. Lower L0 likely pushes the SAE to absorb dense features like spelling information, increasing feature sparsity. Feature absorption rate vs L0 for Gemma Scope SAEs layers 0-17 is shown in Figure 9b. Absorption rate by letter is shown in Appendix A.14. We also train our own set of standard L1 loss SAEs on the first 8 layers of Qwen2 0.5B [32] and Llama 3.2 1B [6], and TopK SAEs [10] on Llama 3.2 1B. In Figure 9c we show that absorption occurs in these SAEs as well.

Our metric cannot capture absorption past layer 17 in Gemma 2 2B since we rely on ablation experiments to be certain the absorbed feature causally mediates model behavior. Past layer 17, attention has already moved the starting letter information from the source token into the final token position, so any ablations on the source token past layer 17 have little effect. This is a limitation of our absorption metric - we rely on ablation to be certain of the causal impact of absorbed features on model behavior, but this limits the layer depth our metric can be applied. We discuss this further in Appendix A.12 and discuss alternative formulations of the metric in Appendix A.13.

Our absorption metric is not perfect, and is likely an under-estimate of the true level of feature absorption. We only consider absorption to have occurred if a single SAE latent has a much larger ablation effect than all other latents, and if the main SAE latents for a feature do not activate at all. Our metric will not capture multiple absorbing latents activating together, or the main latents activating weakly. Regardless, we feel our metric is a reasonable conservative baseline.

## 6 Related work

**Applications of Probes and SAEs for Model Interpretability** Probing methods can extract interpretable information from language models, though this does not guarantee the model uses these representations in its computation, and requires labeled data [7].

Prior work has shown that many human-interpretable concepts in LLM activations are represented as linear directions in activation space, known as the linear representation hypothesis [8, 28]. Li et al. [18] used non-linear probes to recover board representations from a transformer trained on Othello scripts ("OthelloGPT"). Nanda [23] later showed that linear representations were not only recoverable but also editable.

Karvonen et al. [16] developed objective metrics for SAE evaluation using Chess and Othello board states, but does not apply these to SAEs trained on LLM activations. Work by Olah et al. [26], Kissane et al. [17], Templeton et al. [31] noted poor precision/recall of SAE latents compared to known proxies. We extend this by showing how sparsity mediates precision/recall across many Gemma Scope SAEs and offer a possible explanation of low recall due to feature absorption.

Engels et al. [9] investigated SAE errors, finding that not all SAE error is linearly decomposable.

**Studying precision and recall of SAE Latents**   Most existing work on SAE interpretability mainly studies max activating examples [5], which may be misleading. There are more rigorous works which only measure precision [2, 31, 17]. Recent work has briefly explored recall and found it to be worse than expected naively, but this remains poorly understood [26]. We build on this work by evaluating precision / recall on a large number of SAEs, and offer a partial explanation for lower-than-expected recall of SAE latents in the form of feature absorption.

**Decomposing SAE Latents**   Feature splitting was first described in Bricken et al. [2], which noted that different SAE widths and sparsities induce latents of different granularity, with wider SAEs often learning more specific variants of features. Bussmann et al. [4] find that by training an SAE on the decoder of another SAE, a technique called Meta-SAEs, it is possible to break down a single SAE latent like "Einstein" into subcomponents like "German" and "Physicist" and "starts with E".

# 7   Discussion

**Limitations**   Our Absoption metric uses ablation effect to ensure that the absorbed features causally mediate model behavior, and thus might not be easily transferable to the final model layers. Alternate metric formulations mitigating this are discussed in Appendix A.13. Due to compute constraints, we only train and evaluate a small number of non-JumpReLU SAEs in Figure 9. As our goal was only to show absorption occurs in all SAE architectures, we did not feel this is a significant drawback.

**Future Work**   The primary goal of future work is to find solutions to feature absorption. We are particularly hopeful that work extending Meta-SAEs [4] may solve or mitigate feature absorption. Another possible solution may be attribution dictionary learning [25]. Finally, structured sparsity techniques such as group lasso [13] or hierarchical sparse coding [14] may also be a promising direction of future work.

Other possible directions include allowing absorption to occur and using it as a way to recover hierarchies between features in a LLM. Our toy model results suggest that absorption leads to an asymmetric pattern in the encoder and decoder of the SAE, so it may be possible to use this insight to detect absorption (although there may be other reasons for an asymmetry in the SAE encoder and decoder beyond absorption).

**Conclusion**   We identify a form of feature splitting we call "feature absorption", where more specific latents "steal credit" from more general ones. Absorption creates an interpretability illusion, where a seemingly interpretable latent has arbitrary false negatives in its mainline interpretation. Lower recall poses problems for using SAEs for high-stakes classification or finding sparse circuits [21], as the number of latents needed to characterize model behavior may be much larger than expected.

We show that absorption is a consequence of hierarchical co-occurrence between sparse and dense features. If a dense feature like "starts with letter D" always co-occurs with a more sparse feature like "dogs", the SAE can increase sparsity by absorbing the "starts with D" feature into a "dogs" latent.

We hope that our work highlights the fundamental limitations of sparse feature extraction and prompts future research on SAEs such as identifying cases where a feature "should have activated" but does not due to absorption, and exploring theoretical solutions to absorption. The ease of demonstrating absorption in toy models makes it easier to validate potential solutions.

## Acknowledgments and Disclosure of Funding

This project was produced as part of the LASR Labs research program. DC was supported thanks to EPSRC EP/S021566/1.

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

# A Appendix

## A.1 Glossary of Terms

**Sparse Autoencoders (SAEs):** Neural networks trained to reconstruct their input while enforcing sparsity in their hidden layer. In the context of this paper, SAEs are used to decompose the dense activations of language models into more interpretable features.

**SAE error term:** When inserting a SAE into the computation path of the model, errors in SAE reconstruction will propagate to later parts of the model and can change the model output. We refer to the error as the SAE error term, and corresponds to the difference between the SAE output and the original SAE input activation. Marks et al. [21] introduced the idea of adding this error term back to the SAE output to ensure that the SAE does not change model output.

**Latent:** We refer to neurons in the hidden layer of a SAE as latents to avoid overloading the term "feature". This is in contrast to earlier work which used the term "feature" to refer to both human-interpretable concepts and SAE hidden layer neurons.

**Feature:** We use the term "feature" to refer to an idealized human-interpretable concept that the model represent in its activations and which a SAE latent may or may not represent.

**Monosemantic:** Referring to a feature or representation that corresponds to a single, clear semantic concept. In the context of SAEs, a monosemantic feature would ideally capture one interpretable aspect of the input.

**Interpretable:** A latent being interpretable is not well defined in the field, making it difficult to ensure that different authors mean the same thing when referring to SAE interpretability. When we refer to a SAE latent as being interpretable in this work, we mean that it should behave in line with how it appears to behave after inspecting its activation patterns. If an SAE latent appears to track a feature X by a reasonable inspection of its activations but has subtle deviations from this behavior in reality, we say this is not interpretable. We thus measure interpretability via classification performance when a latent appears to be a classifier over some feature.

**Feature dashboard:** A dashboard showing activation patterns and max-activating examples for a SAE latent. Feature dashboards are commonly used to interpret the behavior of an SAE latent.

**Neuronpedia:** A platform, `https://neuronpedia.org`, which hosts feature dashboards for popular SAEs [20].

**Token-aligned latent:** A latent which seems to roughly fire on variants of the same token. For instance, a "Snake" token-aligned latent may fire on the tokens "Snake", "SNAKE", "_snakes", etc...

**Feature splitting:** A phenomenon in SAEs introduced by Bricken et al. [2], where a SAE latent tracks a general feature in a narrow SAE, but splits into multiple more specific SAE latents in a wider SAE. For instance, a latent tracking "starts with L" in a narrow SAE may split into a latent tracking "starts with capital L" and a latent tracking "starts with lowercase L" in a wider SAE.

**Feature absorption:** A problematic form of feature splitting where a SAE latent appears to track an interpretable feature, but that latent has seemingly arbitrary exception cases where it fails to fire. Instead, a more specific latent "absorbs" the feature direction and fires in place of the main latent.

**Circuit:** In the context of neural network interpretability, a circuit refers to a subgraph of neurons or latents within a neural network that work together to perform a specific function or computation. The study of circuits aims to understand how different components of a neural network interact to process information and produce outputs.

**Linear probe:** A simple linear classifier (typically logistic regression) trained on the hidden activations of a neural network to predict some property or task. Used to assess what information is linearly decodable from the network's representations.

**K-sparse probing:** A variant of linear probing where only the k most important latents (as determined by some selection method) are used to train the probe. This helps identify which specific neurons or latents are most relevant for a given task.

**Ablation study:** An experimental method where a component of a system (in this case, a neuron or latent in a neural network) is removed or altered to observe its effect on the system's performance. This helps determine the causal importance of the component.

**Integrated gradients (IG):** An attribution method that assigns importance scores to input latents by accumulating gradients along a path from a baseline input to the actual input. In this paper, it's used as an approximation technique for ablation studies.

**In-context learning (ICL):** A paradigm where a language model is given examples of a task within its input prompt, allowing it to adapt to new tasks without fine-tuning. Often used with few-shot learning techniques.

**Residual stream:** In the context of transformer architectures, the residual stream refers to the main information flow that bypasses the self-attention and feed-forward layers through residual connections.

**Logits:** The raw, unnormalized outputs of a neural network's final layer, before any activation function (like softmax) is applied. In language models, logits typically represent the model's scores for each token in the vocabulary.

**Activation patching:** An interpretability technique where activations at specific locations in a neural network are replaced or modified to observe the effect on the network's output. This helps in understanding the causal role of different parts of the network in producing its final output.

## A.2 Proof: absorption decreases SAE loss for hierarchical features

We analyze the effect of a specific form of feature absorption, termed $\delta$-absorption, within a Sparse Autoencoder (SAE) framework. We consider two hierarchically related features, $f_1$ and $f_2$ (where $f_2 \subset f_1$), and demonstrate that for a defined family of encoder and decoder weights parameterized by $\delta \in [0, 1]$ ($\delta = 0$ corresponds to no absorption, and $\delta = 1$ corresponds to full absorption):

1. Perfect reconstruction of inputs composed of these features is maintained across all values of $\delta$.

2. The sparsity loss component attributable to these features is a decreasing function of $\delta$, provided the child feature $f_2$ has a non-zero probability of appearing.

3. Consequently, optimizing for sparsity encourages higher values of $\delta$, i.e., greater absorption.

## 1. Preliminaries and Assumptions

**H1. Dataset and Features** Let $\mathcal{D}$ be a dataset. We consider a set of features $\mathcal{F} = \{f_1, f_2, \ldots, f_d\}$. Each feature $f_i \in \mathbb{R}^k$ is a vector with unit norm, $\|f_i\|_2 = 1$. Features are mutually orthogonal: $f_i \cdot f_j = \delta_{ij}$ (Kronecker delta), where $\delta_{ij} = 1$ if $i = j$ and $0$ if $i \neq j$. An activation $h \in \mathbb{R}^k$ in the model's residual stream is a linear combination of active features: $h = \sum_{j \in \text{ActiveFeatures}} f_j$.

**H2. Feature Hierarchy and Probabilities** We focus on two features $f_1, f_2 \in \mathcal{F}$ with a hierarchy $f_2 \subset f_1$. This implies that if $f_2$ is present in a datapoint, $f_1$ must also be present. The probabilities of observing combinations of $f_1$ and $f_2$ are:

- $p(f_1, f_2) = p_{11}$: Probability of $f_1$ and $f_2$ co-occurring (e.g., input is $f_1 + f_2$).
- $p(f_1, \neg f_2) = p_{10}$: Probability of $f_1$ occurring without $f_2$ (e.g., input is $f_1$).
- $p(\neg f_1, f_2) = p_{01}$: Probability of $f_2$ occurring without $f_1$. By the hierarchy assumption, $p_{01} = 0$.
- $p(\neg f_1, \neg f_2) = p_{00}$: Probability of neither $f_1$ nor $f_2$ occurring (e.g., input is $0$ or some $f_k, k \neq 1, 2$).

We assume $p_{11} + p_{10} + p_{00} = 1$.

**H3. Sparse Autoencoder (SAE) Model** The SAE reconstructs an input $h$ as $\hat{h} = f_\phi(h)$. The reconstruction is $\hat{h} = W_d z$, where $z = \text{ReLU}(W_e h)$. No bias terms are used. $W_{e,i}$ is the $i$-th row of $W_e$ (encoder vector for latent $i$), and $W_{d,i}$ is the $i$-th column of $W_d$ (decoder vector for latent $i$). We analyze two specific latents, $z_1$ and $z_2$, intended to capture $f_1$ and $f_2$. Other latents $z_j$ for $j > 2$ are assumed to perfectly reconstruct other features $f_j$ (e.g. $W_{e,j} = f_j, W_{d,j} = f_j$) and do not interact with $f_1, f_2$ due to orthogonality.

**H4. SAE Loss Function**    The total loss is $\mathcal{L} = \mathcal{L}_{\text{rec}} + \lambda \mathcal{L}_{\text{sp}}$, where $\lambda > 0$. $\mathcal{L}_{\text{rec}} = \mathbb{E}_{h \sim \mathcal{D}} \left[ \|h - \hat{h}\|_2^2 \right]$.
$\mathcal{L}_{\text{sp}} = \mathbb{E}_{h \sim \mathcal{D}} \sum_i |z_i|$. Our analysis will focus on the contributions of $z_1, z_2$ to $\mathcal{L}_{\text{rec}}$ and $\mathcal{L}_{\text{sp}}$.

**H5. Definition of $\delta$-Absorption**    We define a specific parameterization for the encoder and decoder weights associated with $f_1$ and $f_2$ by a parameter $\delta \in [0, 1]$:

- $W_{e,1} = f_1 - \delta f_2$
- $W_{e,2} = f_2$
- $W_{d,1} = f_1$
- $W_{d,2} = f_2 + \delta f_1$

$\delta = 0$ represents no absorption, while $\delta = 1$ represents full absorption.

## 2. Proposition 1: Perfect Reconstruction under $\delta$-Absorption

*For any $\delta \in [0, 1]$, and for inputs $h$ consisting only of $f_1$, $f_2$ or 0, the reconstruction $\hat{h} = W_{d,1}z_1 + W_{d,2}z_2$ perfectly reconstructs $h$, i.e., the reconstruction loss component $\mathcal{L}_{rec}^{(1,2)}$ due to these features is 0.*

**Proof**    We consider the possible input types based on $f_1, f_2$:

**Case 1: $h = f_1$ (only parent feature $f_1$ is present).**    The latent activations are:
$$\begin{aligned}
z_1 &= \text{ReLU}(W_{e,1} \cdot h) = \text{ReLU}((f_1 - \delta f_2) \cdot f_1) \\
&= \text{ReLU}(f_1 \cdot f_1 - \delta f_2 \cdot f_1) \\
&= \text{ReLU}(1 - \delta \cdot 0) = 1 \quad \text{(by H1)} \\
z_2 &= \text{ReLU}(W_{e,2} \cdot h) = \text{ReLU}(f_2 \cdot f_1) = \text{ReLU}(0) = 0 \quad \text{(by H1)}
\end{aligned}$$
The reconstruction is:
$$\hat{h} = z_1 W_{d,1} + z_2 W_{d,2} = 1 \cdot f_1 + 0 \cdot (f_2 + \delta f_1) = f_1$$

Thus, $\hat{h} = h$.

**Case 2: $h = f_1 + f_2$ (both parent $f_1$ and child $f_2$ are present).**    The latent activations are:
$$\begin{aligned}
z_1 &= \text{ReLU}(W_{e,1} \cdot h) = \text{ReLU}((f_1 - \delta f_2) \cdot (f_1 + f_2)) \\
&= \text{ReLU}(f_1 \cdot f_1 + f_1 \cdot f_2 - \delta f_2 \cdot f_1 - \delta f_2 \cdot f_2) \\
&= \text{ReLU}(1 + 0 - \delta \cdot 0 - \delta \cdot 1) = \text{ReLU}(1 - \delta) \quad \text{(by H1)}
\end{aligned}$$
Since $\delta \in [0, 1]$, $1 - \delta \geq 0$, so $z_1 = 1 - \delta$.
$$\begin{aligned}
z_2 &= \text{ReLU}(W_{e,2} \cdot h) = \text{ReLU}(f_2 \cdot (f_1 + f_2)) \\
&= \text{ReLU}(f_2 \cdot f_1 + f_2 \cdot f_2) \\
&= \text{ReLU}(0 + 1) = 1 \quad \text{(by H1)}
\end{aligned}$$
The reconstruction is:
$$\begin{aligned}
\hat{h} &= z_1 W_{d,1} + z_2 W_{d,2} = (1 - \delta)f_1 + 1 \cdot (f_2 + \delta f_1) \\
&= (1 - \delta)f_1 + f_2 + \delta f_1 = f_1 - \delta f_1 + f_2 + \delta f_1 = f_1 + f_2
\end{aligned}$$

Thus, $\hat{h} = h$.

**Case 3: $h = 0$ (neither $f_1$ nor $f_2$ is present).**
$$\begin{aligned}
z_1 &= \text{ReLU}((f_1 - \delta f_2) \cdot 0) = 0 \\
z_2 &= \text{ReLU}(f_2 \cdot 0) = 0
\end{aligned}$$
The reconstruction is:
$$\hat{h} = 0 \cdot f_1 + 0 \cdot (f_2 + \delta f_1) = 0$$

Thus, $\hat{h} = h$.

**Case 4:** $h = f_2$ **(only child feature $f_2$ is present).** This case is disallowed by assumption H2 ($p_{01} = 0$), as $f_2 \subset f_1$ implies $f_1$ must be present if $f_2$ is.

In all permissible cases, $h - \hat{h} = 0$, so $\|h - \hat{h}\|_2^2 = 0$. Therefore, the reconstruction loss component due to $f_1, f_2$, denoted $\mathcal{L}_{\text{rec}}^{(1,2)}$, is 0 for any $\delta \in [0, 1]$.

## 3. Proposition 2: Sparsity Loss under $\delta$-Absorption

*The expected sparsity loss contribution from latents $z_1$ and $z_2$, denoted $\mathcal{L}_{sp}^{(1,2)} = \mathbb{E}_{h \sim \mathcal{D}}[|z_1| + |z_2|]$, is given by:*

$$\mathcal{L}_{sp}^{(1,2)} = p_{11}(2 - \delta) + p_{10}$$

*Furthermore, its derivative with respect to $\delta$ is:*

$$\frac{d\mathcal{L}_{sp}^{(1,2)}}{d\delta} = -p_{11}$$

**Proof**  We calculate the sum of absolute latent activations $|z_1| + |z_2|$ for each case from Proposition 1 and weight them by their probabilities (H2):

- If $h = f_1 + f_2$ (probability $p_{11}$): $z_1 = 1 - \delta$, $z_2 = 1$. Since $\delta \in [0, 1]$, $1 - \delta \geq 0$, so $|z_1| = 1 - \delta$. $|z_2| = 1$. Thus, $|z_1| + |z_2| = (1 - \delta) + 1 = 2 - \delta$.

- If $h = f_1$ (probability $p_{10}$): $z_1 = 1$, $z_2 = 0$. Thus, $|z_1| + |z_2| = 1 + 0 = 1$.

- If $h = 0$ (probability $p_{00}$, neither $f_1$ nor $f_2$ present): $z_1 = 0$, $z_2 = 0$. Thus, $|z_1| + |z_2| = 0 + 0 = 0$.

The case corresponding to $p_{01}$ does not occur.

The expected sparsity loss from $z_1, z_2$ is:

$$\begin{aligned}
\mathcal{L}_{\text{sp}}^{(1,2)} &= p_{11} \cdot (2 - \delta) + p_{10} \cdot 1 + p_{00} \cdot 0 \\
&= p_{11}(2 - \delta) + p_{10}
\end{aligned}$$

Taking the derivative with respect to $\delta$:

$$\frac{d\mathcal{L}_{\text{sp}}^{(1,2)}}{d\delta} = \frac{d}{d\delta}(2p_{11} - \delta p_{11} + p_{10})$$

Since $p_{11}$ and $p_{10}$ are constants with respect to $\delta$:

$$\frac{d\mathcal{L}_{\text{sp}}^{(1,2)}}{d\delta} = -p_{11}$$

$\square$

## 4. Corollary: Increasing Absorption Decreases Sparsity Loss

*If $p_{11} > 0$ (i.e., the child feature $f_2$ co-occurs with $f_1$ with non-zero probability), then increasing $\delta$ strictly decreases $\mathcal{L}_{sp}^{(1,2)}$.*

**Proof**  From Proposition 2, $\frac{d\mathcal{L}_{\text{sp}}^{(1,2)}}{d\delta} = -p_{11}$. If $p_{11} > 0$, then $-p_{11} < 0$. A negative derivative implies that $\mathcal{L}_{\text{sp}}^{(1,2)}$ is a decreasing function of $\delta$ for $\delta \in [0, 1]$. The minimum value of $\mathcal{L}_{\text{sp}}^{(1,2)}$ over this interval occurs at $\delta = 1$ (full absorption), yielding $\mathcal{L}_{\text{sp}}^{(1,2)}(\delta = 1) = p_{11}(2 - 1) + p_{10} = p_{11} + p_{10}$. The maximum value occurs at $\delta = 0$ (no absorption), yielding $\mathcal{L}_{\text{sp}}^{(1,2)}(\delta = 0) = p_{11}(2 - 0) + p_{10} = 2p_{11} + p_{10}$. $\square$

## 5. Conclusion

Given the specified $\delta$-absorption mechanism for an SAE handling two hierarchical features $f_1, f_2$ (where $f_2 \subset f_1$):

1. Perfect reconstruction of inputs composed of $f_1$ and $f_2$ is maintained irrespective of the degree of absorption $\delta$. Thus, $\mathcal{L}_{\text{rec}}^{(1,2)}$ is unaffected by $\delta$.

2. The sparsity loss component $\mathcal{L}_{\text{sp}}^{(1,2)}$ associated with these features is $p_{11}(2 - \delta) + p_{10}$.

3. If $p_{11} > 0$, the total loss $\mathcal{L}$ (focusing on the components related to $f_1, f_2$) decreases as $\delta$ increases because $\mathcal{L}_{\text{rec}}^{(1,2)}$ is constant (zero) and $\mathcal{L}_{\text{sp}}^{(1,2)}$ decreases.

4. Therefore, an optimization process like gradient descent, when minimizing the total loss $\mathcal{L} = \mathcal{L}_{\text{rec}} + \lambda\mathcal{L}_{\text{sp}}$ (where $\lambda > 0$), will favor increasing $\delta$ towards 1, thereby promoting feature absorption for these hierarchically related features, assuming the SAE learns or is constrained to these forms of $W_e$ and $W_d$.

This formalizes the argument that, under the given conditions and definitions, absorption is a mechanism that can reduce SAE loss by improving sparsity without harming reconstruction for hierarchical features.

### A.3  Extended toy model experiments

In this section we explore further variants on absorption in toy models. We use the same setting as our main toy model experiment, with four mutually-orthogonal true features, and train an SAE with four latents. Each true feature $f \in \mathbb{R}^{50}$. Unless otherwise stated, every time feature 1 fires feature 0 must also fire, but feature 0 is allowed to fire on its own as well. This is to simulate hierarchal features such as our example "starts with S" and "short" features, where every time the "short" feature fires we expect "starts with S" must also fire since "short" starts with "S", but "starts with S" can fire on its own as well. Feature 2 and 3 are fully independent. All features fire with magnitude 1.0 and variance 0.0 unless otherwise stated.

**Magnitude variance causes partial absorption**  In our main toy model experiment, each true feature fires with magnitude exactly 1.0. This is not very realistic, though - likely there will be some variance in feature firing magnitudes in real LLMs. We simulate this by adding variance of 0.1 to the firing magnitude of feature 0, so the relative magnitudes of feature 0 and 1 are no longer fixed. We show the plots of cosine similarity between SAE encoder and decoder in Figure 10. Here, we still see the same absorption pattern in the SAE encoder and decoder with the latent 3 encoder containing a negative component of feature 1, and the latent 0 decoder merging features 0 and 1. We show some sample true feature firings and corresponding SAE latent activations in Table 2.

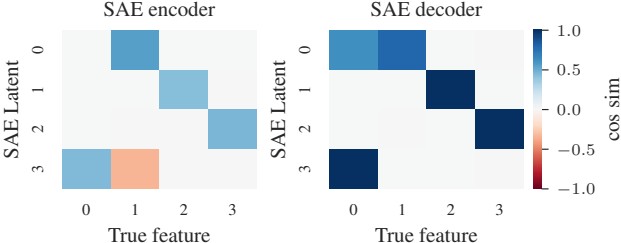

Figure 10: SAE encoder and decoder with true features. The firing magnitude of feature 0 has variance 0.1, while the remaining features fire with variance 0.0. When there is variance in the firing magnitudes of parent and child features, we still see an absorption pattern in the SAE encoder and decoder with the latent 3 encoder containing a negative component of feature 1, and the latent 0 decoder merging features 0 and 1.

We see that now the SAE latent tracking feature 0 still fires when the true values of features 0 and 1 are both 1.0, but very weakly. However, if the magnitude of feature 0 drops down to 0.75, then the feature 0 latent fully turns off.

Table 2: Sample true feature firings and corresponding SAE latent activations. Feature 1 only fires if feature 0 fires. Feature 0 has variance of 0.1 in its firing magnitude, while the other feature have no variance in their firing magnitude.

| TRUE FEATURES | | | | SAE LATENT ACTS | | | |
|---|---|---|---|---|---|---|---|
| **1.00** | 0.00 | 0.00 | 0.00 | 0.00 | 0.00 | 0.00 | **1.00** |
| **1.00** | **1.00** | 0.00 | 0.00 | **1.27** | 0.00 | 0.00 | **0.22** |
| **0.90** | **1.00** | 0.00 | 0.00 | **1.27** | 0.00 | 0.00 | **0.12** |
| **0.75** | **1.00** | 0.00 | 0.00 | **1.26** | 0.00 | 0.00 | 0.00 |

We call this phenonemon **partial absorption**. In partial absorption, there's co-occurrence between a dense and sparse feature, and the sparse feature absorbs the direction of the dense feature. However, the SAE latent tracking the dense feature still fires when both the dense and sparse feature are active, only very weakly. If the magnitude of the dense feature drops below some threshold, it stops firing entirely.

Feature absorption is an optimal strategy for minimizing the L1 loss and maximizing sparsity. However, when a SAE absorbs one latent into another, the absorbing latent loses the ability to modulate the magnitudes of the underlying features relative to each other. The SAE can address this by firing the latent tracking the dense feature as a "correction" to add back some of the dense feature direction into the reconstruction. Since the dense feature latent is firing weakly, it still has lower L1 loss than if the SAE fully separated out the features into their own latents.

**Imperfect co-occurrence can still lead to partial absorption**    Next, we test what will happen if feature 1 is more likely to fire if feature 0 is active, but can still fire without feature 0. We set up feature 1 to fire with feature 0 95% of the time, but 5% of the time it can fire on its own. For this experiment, all features fire with magnitude 1.0 and 0 variance. We show the cosine similarities of the SAE encoder and decoder with true features in Figure 11. Some sample feature firings and corresponding SAE activations are shown in Table 3.

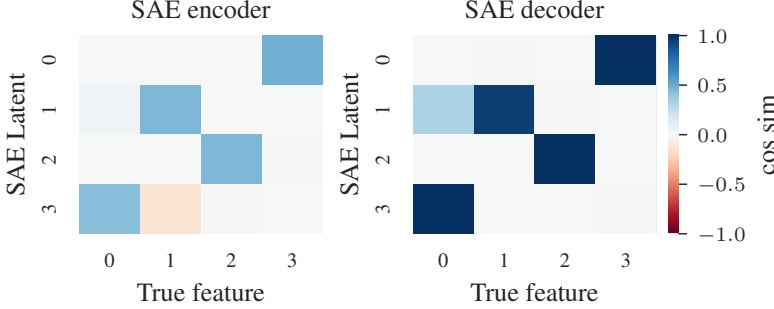

Figure 11: SAE encoder and decoder with true features. Feature 1 fires with feature 0 95% of the time, but 5% of the time feature 1 is allowed to fire on its own. Even though there is not perfect co-occurrence between features 0 and 1, we still see the SAE encoder and decoder learn a weak absorption pattern, with the decoder latent for feature 1 absorbing part of feature 0, and the encoder latent for feature 0 including a negative component of feature 1.

We see signs of partial absorption here as well. We see the same absorption pattern in the SAE encoder and decoder as we saw in our other absorption examples, although less severe than the previous examples. We also see in the sample firing patterns that when both feature 0 and 1 fire together, the latent tracking feature 0 fires with noticeably lower magnitude than when feature 0 fires on its own. Here, even though the co-occurrence between features 0 and 1 is not perfect, we still see partial absorption.

**Absorption also affects TopK SAEs**    So far, we have only shown feature absorption occurring with standard L1 SAEs. Next, we examine how other absorption affects other architectures using

Table 3: Sample feature values and corresponding SAE activations. Feature 1 can only fire if feature 0 is active 95% of the time, but 5% of the time feature 1 can fire on its own. We see signs of partial absorption, where the latent tracking feature 0 fires noticeably more weakly if feature 1 is active.

| TRUE FEATURES | | | | SAE LATENT ACTS | | | |
|---|---|---|---|---|---|---|---|
| **1.00** | 0.00 | 0.00 | 0.00 | 0.00 | 0.00 | 0.00 | **1.00** |
| **1.00** | **1.00** | 0.00 | 0.00 | 0.00 | **1.05** | 0.00 | **0.67** |
| 0.00 | **1.00** | 0.00 | 0.00 | 0.00 | **0.95** | 0.00 | 0.00 |

a batch topk SAE [3]. Batch topk SAEs are an improved version of topk SAEs [10] where the top $k * B$ latents are used to reconstruct the SAE input, where $B$ is the batch size. As the topk function enforces sparsity, there is no additional L1 loss term.

Topk SAEs are harder to use for very small toy models like our 4-feature toy model above, since if the $k$ is too large relative to the size of the SAE the SAE will not learn correct features. To address this, we use a slightly larger toy model with 12 mutually orthogonal true features. All features fire independently with probability 0.15, except for the first 2 features. Feature 0 is the parent feature in our hierarchy, and fires with probability 0.4. Feature 1 is the child feature, and fires with probability 0.6 only if feature 1 fires, but never fires if feature 1 does not fire. All features fire with magnitude 1.0. We train a batch topk SAE with $k = 2$. We show the cosine similarities of the SAE encoder and decoder with true features in Figure 12.

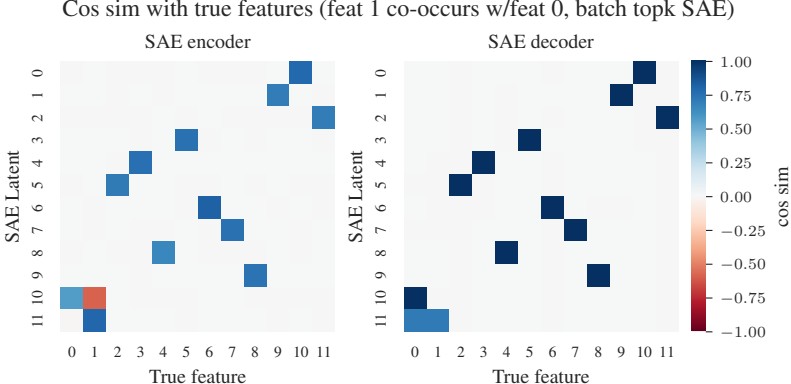

Figure 12: SAE encoder and decoder with true features for a batch topk SAE with k=12. Feature 1 is only allowed to fire if feature 0 fires. We still see a very clear absorption pattern between the latents tracking features 0 and 1 despite the lack of L1 loss.

We still see a clear absorption pattern between the latents tracking features 0 and 1 despite the lack of L1 loss. Absorption increases sparsity, which allows the topk SAE to have better reconstruction loss at a given k, and is thus what the SAE learns.

### A.4 Ablation algorithm

### A.5 How good is Gemma-2 on character identification tasks?

We evaluate how well can Gemma-2-2B identify the first letter or all the letters in a token (spelling the full token). We evaluate the accuracy of the model on all tokens in the LR probe validation set with a prompt containing 10 in-context examples selected at random from the full vocabulary. Our results are shown in Figure 13.

We see that performance on the first-letter identification task is high throughout token length, while the full-word spelling performance decreases as the length of the token increases.

**Algorithm 1** SAE Latent Ablation

Insert SAE in model computation, including error term
Define a scalar metric on the model's output distribution
Calculate baseline metric value for a test prompt
**for** each token of interest **do**
  **for** each SAE latent **do**
    Set the SAE latent activation to 0
    Recalculate the metric
    Compute ablation effect (baseline - new metric)
    Reset the SAE latents to its original value
  **end for**
**end for**=0

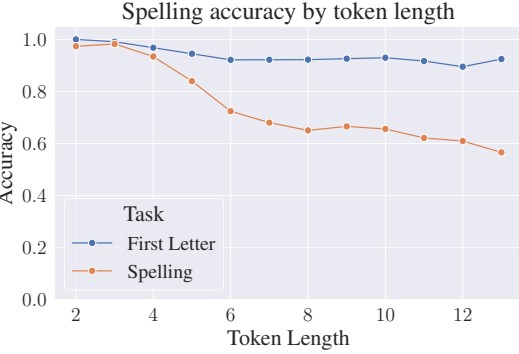

Figure 13: Baseline performance for Gemma-2-2B on first-letter identification and full-token spelling by token length.

## A.6 Intervening on the first letter

If the model is using the identified SAE latents for predicting the first letter we should also be able to change what first letter it predicts just by changing the activations. For this experiment we use the SAE latents most cosine similar with the LR probe for the true first letter and for a new randomly selected letter. We take the intermediate activations of Gemma-2-2B in the residual stream and encode them using the SAE. Then we zero out the activation of the SAE latent associated with the original letter and change the activation of the SAE latent associated with the new letter into the average activation it has on tokens starting with this new letter.

Editing works better with latents from the narrower 16k SAE compared to the 65k, with the best L0s in the 75-150 range. This corresponds to the observed pattern of these SAE latents having higher F1 scores for classification. We report the results in Figure 14. The best SAEs on the layers 7-9 can achieve a substantial replacement, but note that the averages hide variance across individual tokens, where some get edited completely and others get unaffected. The edit success also varies based on the true first letter and the random new letter; for illustration we show a breakdown by letter for two specific SAEs in layer 7 in Figure 15.

## A.7 Probe cosine similarity vs k=1 sparse probing

The first step when searching for a SAE latent that acts as a first-letter classifier involves searching for SAE latent which best acts as a classifier. In Figure 4, we achieve this by first training a LR probe on the first-letter task and using cosine similarity between that probe and the SAE encoder to find the best latent for the first-letter task. We also investigated using k-sparse probing with k=1 to select the best SAE latent instead. This involves training a linear probe with L1 loss and selecting the latent with the highest positive weight from the probe.

We find that both k=1 sparse probing yield nearly identical results, as seen in Figures 16 and 17. Additionally Figure 18 shows the cosine similarity of the LR probe with each SAE latent by letter for the canonical Gemma Scope layer 0 16k width SAE. In most cases there is an obvious probe-aligned

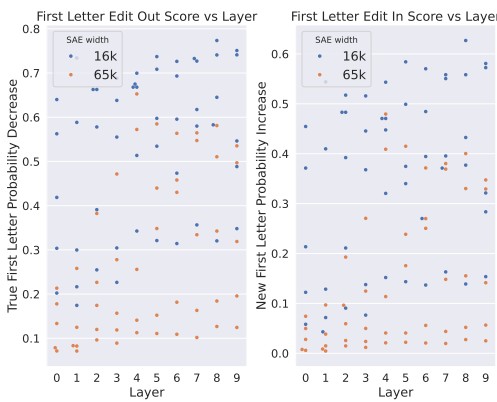 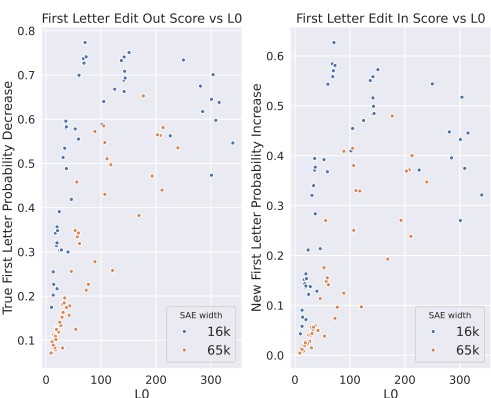

(a) Comparing success in editing out the true first letter and making the model predict a randomly selected new letter across layers 0-9 for all 16k and 65k Gemma Scope SAEs.

(b) Comparing the edit success with the top SAE latent across all L0s for 16k and 65k widths across layers 0-9. The best performance seems to be occurring for L0 between 75 and 150.

Figure 14: Comparison of Edit success by Layer and L0

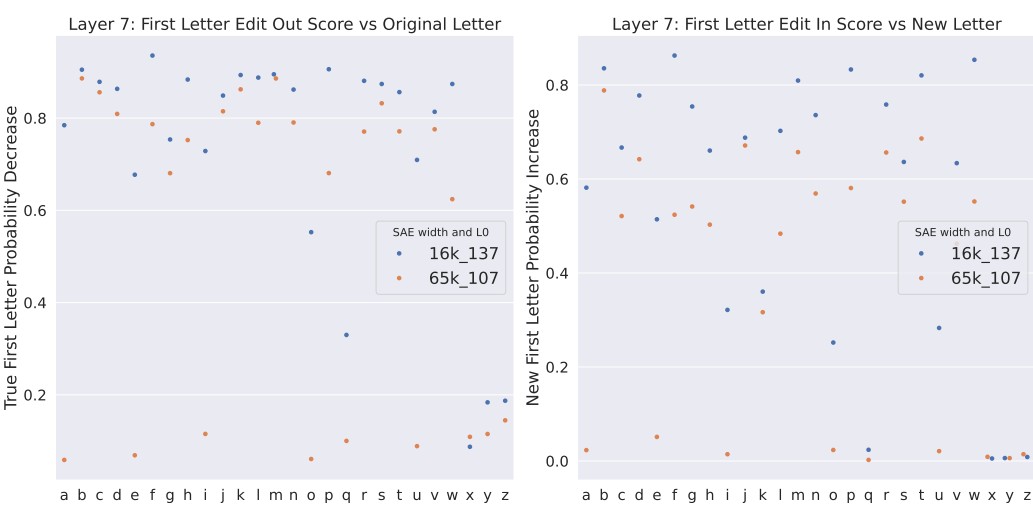

Figure 15: Comparing the edit success broken down by the letter at layer 7 for two SAEs; SAE width 16,000 and L0 of 137 and SAE width 65,000 and L0 of 107. For each original letter we draw a sample of 100 tokens and average the decrease in probability of the correct first letter and increase in probability of a new random letter.

latent. Likely any reasonable method of latent selection will find the same latent for these cases. We thus decided to use cosine similarity between the SAE encoder and a LR probe as our selection criteria for single SAE latents as this is a simpler metric and less computationally intensive to compute.

### A.8 Precision, recall, and F1 score for the first-letter task

We evaluated precision, recall, and F1 score for the first-letter classification task, and found that the precision and recall vary depending on the L0 of the SAE. Low L0 SAEs learn high precision, low recall latents, while high L0 SAEs learn low precision, high recall latents. These results are shown in Figure 19. We thus chose to use F1 score as our core metric in this paper to balance precision and recall as many if the SAEs we tested have extreme values in either precision or recall.

While it may appear that there is an optimal L0 from looking at aggregate statistics across letter, we find that breaking down the F1 vs L0 plot by letter reveals that the optimal L0 appears different for different letters, with low frequency letters like z actually having the best F1 score at the lowest L0,

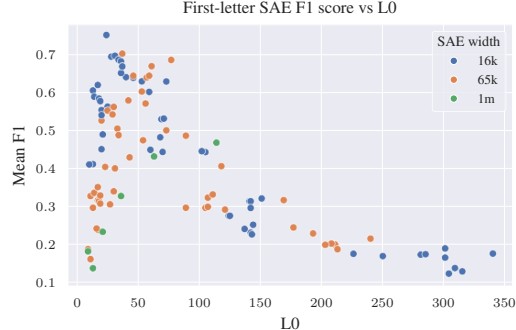
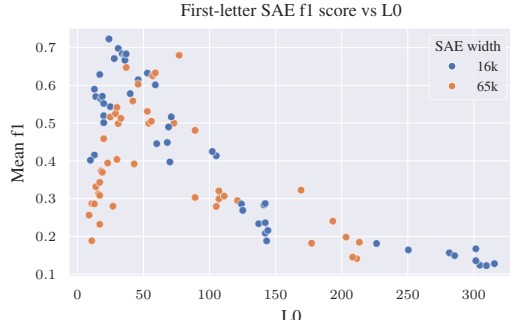

(a) Mean F1 score on first-letter classification task using top SAE encoder latent by cosine similarity with the LR probe vs L0 for all Gemma Scope SAEs layers 0-9.

(b) Mean F1 score on first-letter classification task using k=1 sparse probing to select the SAE latent for the first-letter classification task vs L0 for all Gemma Scope SAEs layers 0-9.

Figure 16: Comparison of LR probe cosine similarity and k=1 sparse probing vs l0

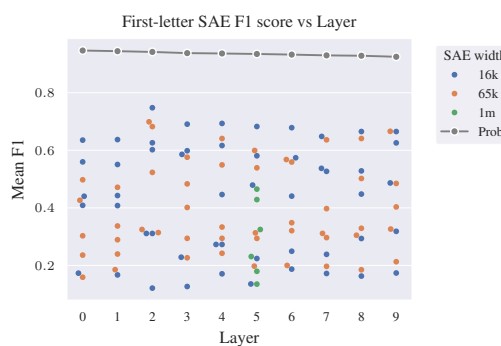
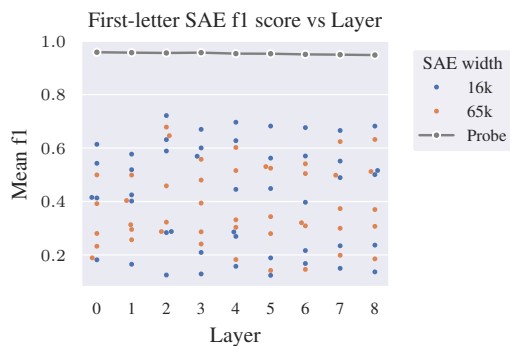

(a) Mean F1 score on first-letter classification task using top SAE encoder latent by cosine similarity with the LR probe vs layer for all Gemma Scope SAEs layers 0-9.

(b) Mean F1 score on first-letter classification task using k=1 sparse probing to select the SAE latent for the first-letter classification task vs layer for all Gemma Scope SAEs layers 0-9.

Figure 17: Comparison of LR probe cosine similarity and k=1 sparse probing vs layer

while other letters instead have an optimal L0 around 30-50. Figure 20 shows these results broken down by letter.

## A.9 SAE training

We train SAEs on the first 8 layers of Qwen2 0.5B [32] and Llama 3.2 1B [6] using the SAELens library [15]. The SAEs are all trained with identical hyperparameters of L1 coefficient of 2.5 and 500M tokens. The Qwen2 0.5B SAEs all have L0 between 25 and 50 and explained variance between 0.77 and 0.83. The Llama 3.2 1B SAEs have L0 between 27 and 110, and explained variance between 0.74 and 0.89. We use a single 40gb Nvidia A100 GPU for training each SAE.

## A.10 Metric choice for ablation studies

To determine the causal effect of SAE latents on the first-letter identification task, we use a metric, $m$, which measures the logit of the correct letter minus the mean logit of all incorrect letters. Our metric is defined below, where $g$ refers to the final token logits, $L$ is the set of uppercase letters, and $y$ is the uppercase letter that is the correct starting letter:

$$m = g[y] - \frac{1}{|L| - 1} \sum_{l \in \{L \setminus y\}} g[l]$$

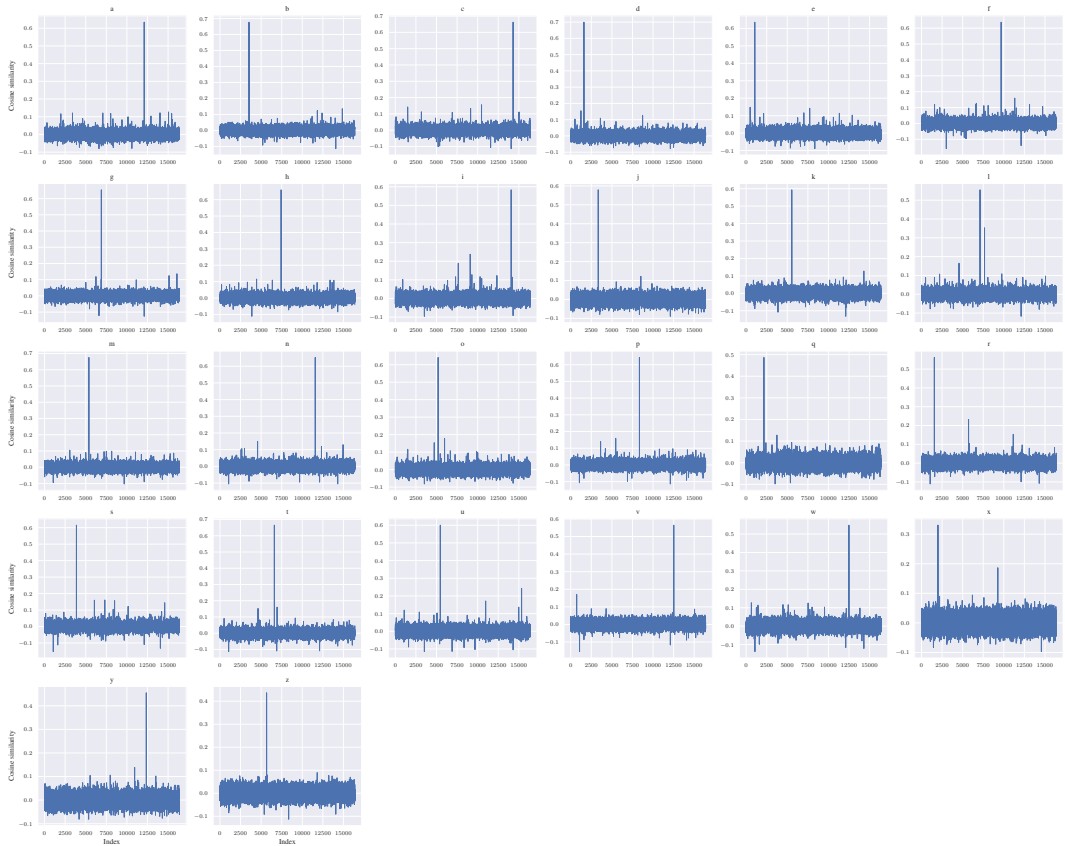

Figure 18: Decoder cosine similarities with the LR probe by letter, Gemma Scope 16k layer 0 l0=105. Most letters have one or two obvious SAE latents which align with the probe.

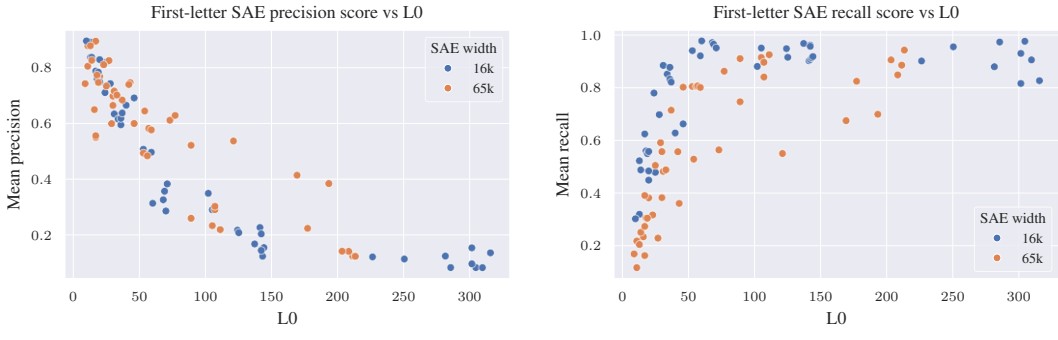

(a) Mean precision on first-letter classification task vs L0 for all Gemma Scope SAEs layers 0-9. Latents are selected via k=1 sparse probing

(b) Mean recall on first-letter classification task vs L0 for all Gemma Scope SAEs layers 0-9. Latents are selected via k=1 sparse probing

Figure 19: Comparison of precision and recall vs l0

This metric is chosen to detect changes in the confidence of the model in predicting the correct letter relative to the mean reference class of other letters. This should capture changes in the model's confidence in predicting the correct logit.

This is not the only metric that could be chosen, and an argument can be made that we should subtract the max of all incorrect letter logits rather than the mean of all incorrect letter logits. The max form of this version of the metric is shown below:

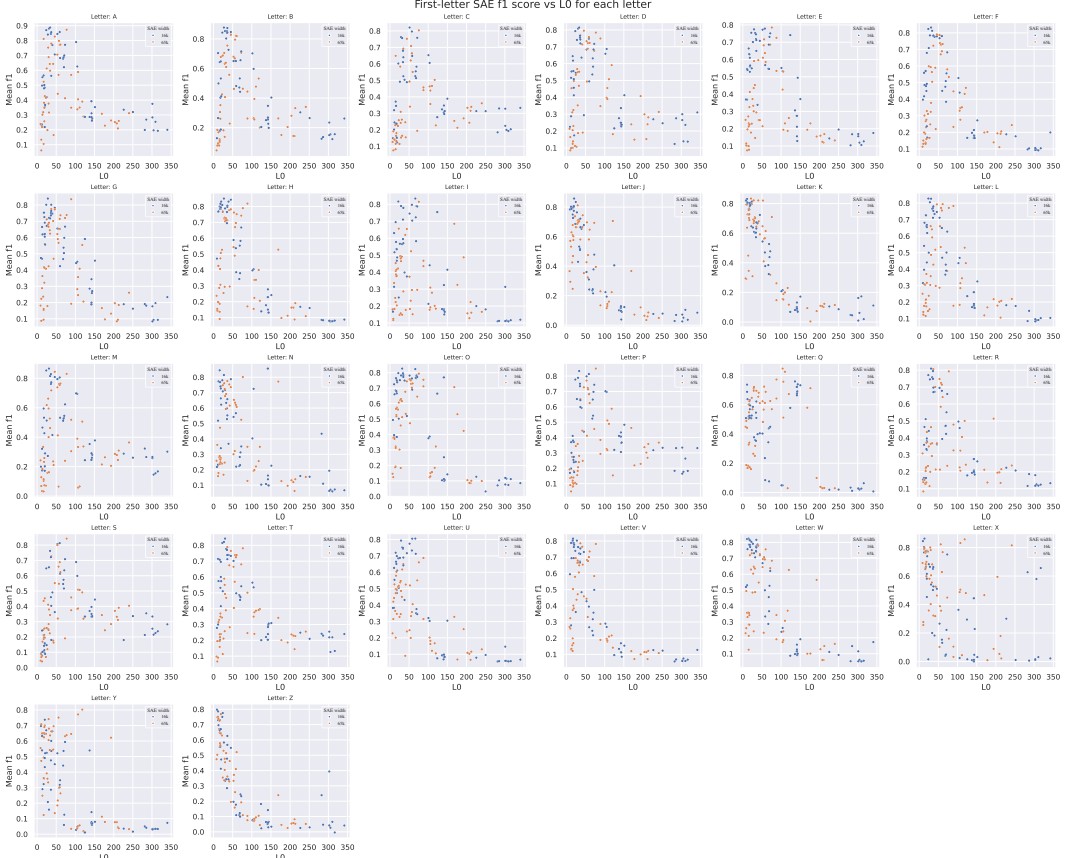

Figure 20: F1 vs L0 by letter. SAE latents are picked using k=1 sparse probing.

$$m_{max} = g[y] - \max_{l \in \{L \setminus y\}} g[l]$$

This second form using a max can also account for the case where the logits of the model shift from being confident in the correct answer to instead being confident in an incorrect answer while leaving the logits of the correct answer the same.

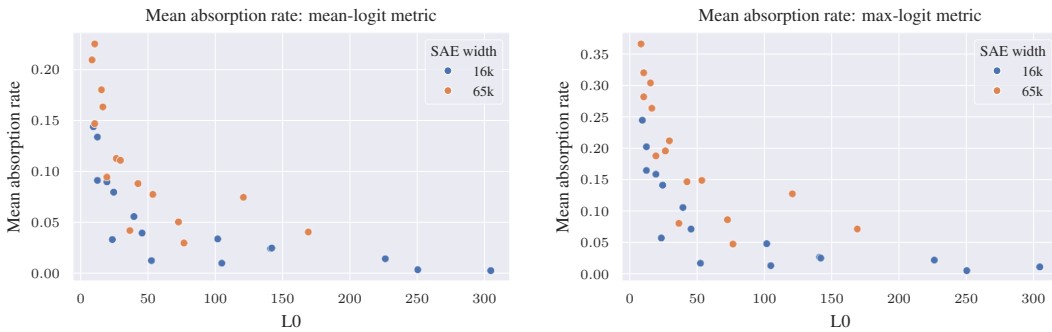

(a) Absorption rate using the mean version of the absorption metric, Gemma Scope layers 0-3.

(b) Absorption rate using the max version of the absorption metric, Gemma Scope layers 0-3.

Figure 21: Comparison of absorption rates using the max and mean versions of the absorption metric.

In practice, we expect that ablating an absorbing latent should cause the model to become less confident in the correct answer, so the difference between these two forms of the metric should yield similar results.

We calculate the mean absorption rate for Gemma Scope SAEs layers 0-3 in Figure 21 using both versions of this metric. The overall shape of the curve is nearly identical between these two choices of metrics. The mean version of the metric, which is used in this paper, results in a slightly more conservative estimate of absorption rate.

We consider our absorption score to be a rough estimate of the true absorption rate and thus consider either the mean or the max version of the logit diff metric to be valid for evaluating absorption.

## A.11 Choice of thresholds for absorption metric

Our absorption metric makes use of several thresholds. For feature splitting, we use a threshold of 0.03 on the F1 score jump moving from $k = n$ to $k = n + 1$ in sparse probing. To classify a latent as a case of absorption, we require cosine similarity with the LR probe above 0.025, and a clear largest ablation effect of 1.0 more than then next highest latent. In this section, we discuss the intuition behind these thresholds. In all cases, these thresholds are just the rough midpoint of ranges of values that achieve similar results, and that we feel are reasonable default values. We also refer readers to our online feature absorption explorer to gain a similar intuition as to what typical ranges of these values look like.

**Feature splitting threshold**   We detect interpretable feature splitting by noting how large a jump in F1-score we gain moving from $k = n$ to $k = n + 1$ in sparse probing. Figure 8a demonstrates a typical example of how F1 score increases for interpretable feature splitting vs feature absorption. In the left side of the figure, we see that moving from $k = 1$ to $k = 2$ there is a F1-score jump of around 0.08, while each increase after that is less than 0.01. For the figure on the right, where no interpretable feature splitting occurs, all F1-score jumps are less than 0.01. This plot is a very typical illustration of detecting feature splitting via k-sparse probing. Any threshold between 0.01 and 0.05 does a good job of detecting feature splitting. We set the threshold to 0.03 to be in the middle of this range.

**LR probe cosine similarity threshold**   We use a threshold of 0.025 on the cosine similarity of the firing latent with the LR probe as part of the metric, ensuring that any latent we classify as absorption must contain some component of the probe direction. This threshold is mainly a cheap way to filter out latents that are obviously not probe-aligned so we can avoid running the more expensive ablation experiments. Nearly any threshold above 0 and below 0.05 should work identically well for this purpose. We choose 0.025 to be in the middle of this range. For most cases of absorption we detect, the absorbing latent has a probe cosine similarity of around 0.05 - 0.15. Figure 6b demonstrates a very typical case of cosine similarity between an absorbing latent and the LR probe, showing cosine similarity of 0.12.

**Absorption effect threshold**   As part of our metric, we assert that any latent we classify as absorption must have the highest ablation effect of all latents, and that lead must be by at least 1.0. As the main goal of the metric in our paper is establishing definitively that absorption exists and affects real LLM SAEs, we focus on the most obvious cases of absorption where a single latent absorbs the parent feature direction in a single activation. This threshold thus serves two purposes: first, to establish the latent has a strong ablation effect, and second, the establish that the effect is dominant over other latents. The metric is not particularly sensitive to the exact choice of threshold we set here, as any dominant ablation effect above around 0.5 is already quite strong. Figure 7 shows a typical ablation effect for a dominant absorbing latent, and has an ablation effect above 6, while all other latents have ablation effects well below 0.5. We pick a threshold of 1.0 here as its a clean number that is well within the range that will work as a threshold. Any threshold between 0.5 and  3.0 should work just as well.

## A.12 Causal interventions and absorption

In this work, we rely on causal interventions like ablation experiments to verify that SAE latents have a causal impact on model behavior. In these experiments for spelling tasks, we set up an ICL prompt to elicit spelling information from the model, for instance the ICL prompt below:

```
tartan has the first letter: T
mirth has the first letter: M
dog has the first letter:
```

In this ICL prompt, we would apply an SAE and train LR probes on the _dog token position, and expect that the model will output the token _D. When we intervene on the _dog, we can track the causal changes to model outputs by applying a metric to the output logits, e.g. checking how our intervention increases or decreases the _D logit relative to other letters.

We use these interventions as part of our absorption metric to ensure that when we claim that "absorption" is occurring, we verify that the absorbing latent has a causal impact on model outputs. This is stronger evidence than only noting a cosine similarity between the absorbing latent, but this means that our absorption metric cannot classify absorption at later model layers.

During a LLM forward pass, the model first collects relevant information on a token in that token position, and attention heads then move relevant information from earlier tokens to later tokens [11, 22]. If we assess ablation effect at layers after which model attention has already pulled relevant information from the subject tokens into the final output token, the ablation effect will be 0. For Gemma 2 2B on the first-letter spelling task, we find this movement of first-letter spelling information occurs around layer 18.

Figure 22 shows an activation patching experiment [22] on a sample first-letter spelling prompt. In this experiment, we see that near layer 18 the model moves first-letter spelling information from the subject token to the prediction token.

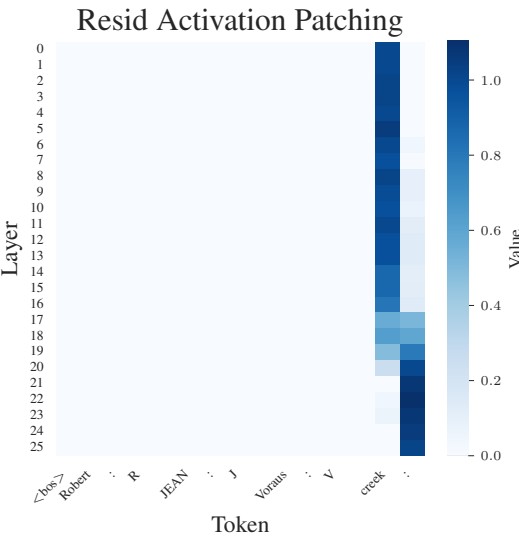

Figure 22: Residual stream attribution patching for a sample first-letter spelling prompt, Gemma 2 2B. After around layer 18, model attention moves the relevant spelling information from the source token to the prediction location.

As a result, our feature absorption metric will not function past layer 18 in Gemma 2 2B, and we thus focus on layers 0-17 for our analysis of feature absorption. We believe that feature absorption is still occurring in SAEs past layer 18, but we lose the ability to make causal claims that the absorbing latents are used by the model to make predictions. Given that this paper is trying to highlight the existence of feature absorption, we felt it is more important to have a metric which is robust and has the backing of causal analysis but which cannot be used at all model layers. Future work may make a different trade-off and choose a feature absorption metric which can work at all model layers, for instance relying only on cosine similarity between absorbing latents and a LR probe to determine absorption. We describe variations on the metric that can facilitate this in Appendix A.13 below.

### A.13 Alternate formulations of the absorption metric

There are a few variations to the metric that can be made to make it more flexible so that it can be applied at all layers of the model, and can be tweaked to detect partial absorption as well. As described above, we did not use this variation as we felt it is more important to demonstrate definitely and conservatively that absorption occurs in this paper. However, we describe the changes that can be made to the metric as follows:

**replace ablation study with LR probe projection in reconstruction** The metric in our paper uses an integrated-gradients ablation study to be absolutely certain that an absorbing feature is causally

responsible for model behavior. We use this in the paper as our goal is to establish with certainty that feature absorption exists in real models, but it has the following drawbacks:

- We need to be able to consistently prompt the model to perform the task, outputting a single token

- We cannot evaluate final model layers after attention moves task-relevant information to the final token

- Ablation studies are slow, which makes the absorption metric expensive to calculate.

The ablation study can be replaced instead with a threshold on the portion of the logistic regression (LR) probe direction an absorbing latent contributes to the residual stream. To do this, we project all firing latents against the LR probe direction $d_p$, as well as project the input activation $a$ against the probe. We require that a latent $l$ must contribute at least $\tau_c$ portion of the probe projection to the reconstruction. So, in order for $l$ to be considered an absorbing latent, the following must be true:

$$ \tau_c < \frac{\hat{a}_l \cdot d_p}{a \cdot d_p} $$

where $\hat{a}_l$ is the reconstruction component of latent $l$ (the encoder activation of latent $l$ times the decoder vector for $l$).

**Allow multiple absorbing latents** The metric in the paper requires one dominant absorbing latent for simplicity, but it is possible in theory to have two or more absorbing latents firing together performing absorption. We can modify the metric to take this into account by allowing up to $N$ latents firing together contribute up to $\tau_c$ together.

Thus, the threshold from change 1 above now becomes the following, where $l_n$ refers to the absorbing latent with the $n$th largest contribution to the LR probe direction in the reconstruction:

$$ \tau_c < \sum_n^N \frac{\hat{a}_{l_n} \cdot d_p}{a \cdot d_p} $$

We do not do this in the paper as it requires another hyper-parameter which is hard to set in a principled way, but should result in a less conservative estimate of absorption.

**Allow main latent(s) to fire weakly instead of being fully turned off** The metric in the paper requires that the main latent(s) for a task all be fully disabled to identify a case of absorption. However, in partial absorption, as explored in our section on toy models, the main latent does not always fully turn off but fires very weakly instead. We can adjust the metric to take this into account by relaxing the requirement that the main latents are fully disabled and instead allow them to fire weakly. Similar to change 1, we define a threshold $\tau_m$ as a maximum contribution to the probe direction in the reconstructed activation $\hat{a}$ that comes from each main latent $l_m$. Thus, if we have $M$ main latents, in order to be classified as absorption the following must be satisfied:

$$ \tau_m \geq \sum_m^M \frac{\hat{a}_{l_m} \cdot d_p}{a \cdot d_p} $$

In the metric defined in the paper, $\tau_m = 0$, meaning the main latents must be fully turned off to count as absorption.

This final change, which detects partial absorption, may be preferable to get a sense of an overall level of absorption present in an SAE. However, if the goal is to determine whether absorption affects the SAE's ability to act as a classifier, then requiring that the main SAE latent be fully turned off to be called absorption is preferable, so we do not claim that either version of the metric is superior in all cases.

## A.14 Additional plots

In this section, we include additional plots that are too large to fit in the main body of the paper.

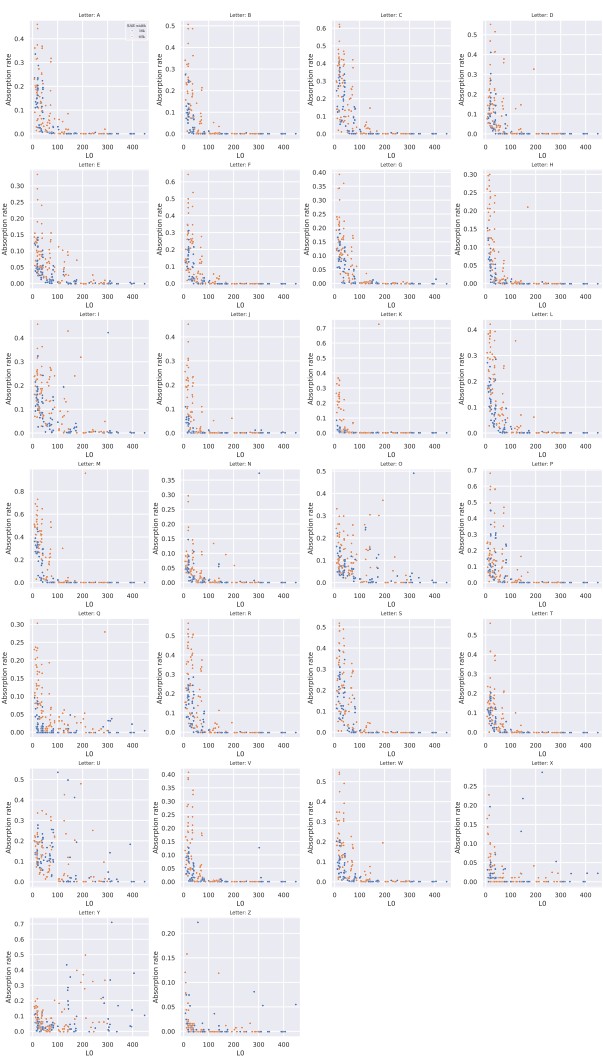

Figure 23: Absorption rate vs L0 by letter, layers 0-17. We see a wide variance in which letters are absorbed by which SAEs.

### A.15 Feature dashboards

We include feature dashboard screenshots from Neuronpedia for some prominent latents mentioned in this work. Figure 24 shows a dashboard for Gemmascope layer 3, latent 1085, which is a token-aligned latent firing on variations of the word `_short` and we find absorbs the "starts with S" direction. Figure 25 shows latent 6510 from the same layer which should be the main "starts with S" latent.

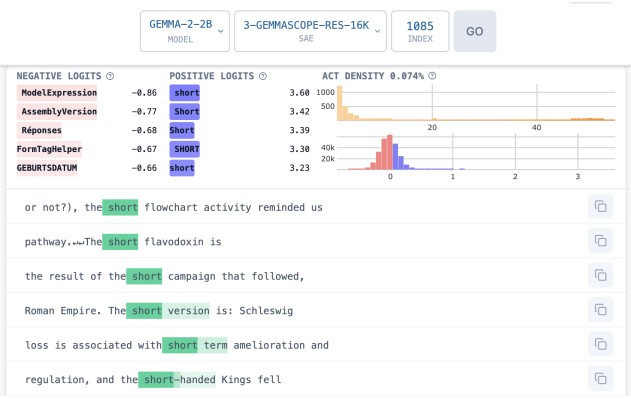

Figure 24: Neuronpedia dashboard for Gemma Scope layer 3, latent 1085. This latent is a token-aligned latent for `_short` tokens. This latent absorbs the "starts with S" direction.

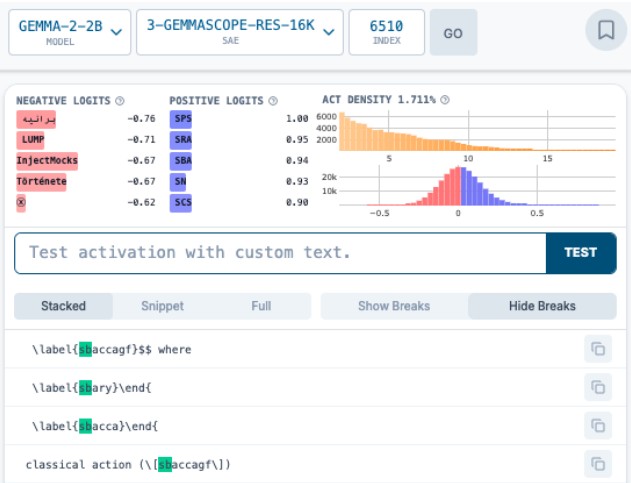

Figure 25: Neuronpedia dashboard for Gemma Scope layer 3, latent 6510. This latent should be the main "starts with S" latent.

