# OpenReview forum: "A is for Absorption: Studying Feature Splitting and Absorption in Sparse Autoencoders"
_NeurIPS.cc/2025/Conference — NeurIPS 2025 oral_

### Official Review · Reviewer_64Na · 2025-06-27

**Clarity:** 3
**Significance:** 2
**Originality:** 3
**Rating:** 4
**Confidence:** 3

**Summary:**

This paper showcases the phenomenon of feature absorption, a mechanism by which sparse autoencoders (SAEs) systematically fail to fully capture features that co-occur in a hierarchical fashion, creating an interpretability illusion. A toy model is used to show that absorbed features are theoretically and empirically preferred by the SAE training procedure. Ablation experiments, cosine similarities to linear probes and k-sparse probes are used to identify specific examples of feature absorption in public base LLMs. A measure of absorption rate is defined and used to quantify the feature absorption in these LLMs.

**Questions:**

1. As I understand (correct me if I am wrong), the main function of the absorption metric is primarily to auto-classify existing feature splits as either benign or harmful to the SAE's ability to detect a known human-interpretable latent. While the prevalence of harmful absorptions is confirmed by the metric, the authors do not evaluate whether these purported absorbed feature events are indeed more harmful towards SAE-derived interpretations than split non-absorbed feature events. I would like to see some evidence as to whether or not this metric is actually a good detector of the quality of an SAE-derived interpretation.

2. My Weakness item #2 comes from my understanding that the purpose of the absorption metric is that it should be used in conjunction with the SAEs when trying to identify the unknown features that may be present in a model. Is this the intended purpose or use case? It is unclear whether the metric is supposed to serve a different purpose instead, e.g., to help with further interpretation of the model after the features have already been identified. If you would explain an example of where you expect this metric to be useful in a different part of the interpretation pipeline, I may increase my score.

minor: I see in the glossary that the authors explain how they avoid overloading the word "feature", which is really helpful for the reader. If I had seen that earlier, reading this paper may have been more straightforward, so I might suggest folding that explanation into the background and also keeping it in the Appendix.

minor: Line 87, capitalization typo on "L1"

**Ethical Concerns:**

["NO or VERY MINOR ethics concerns only"]

**Final Justification:**

The paper convincingly finds that feature absorption occurs within SAEs, stimulating future research into correcting this kind of failure mode. I would give the paper a higher score if it were not for the potential alternate interpretation that the LLMs were truly using separate non-hierarchically co-activating features that were being correctly identified by the SAE, in cases where the authors claim the features are hierarchically co-activating and the SAE found the wrong features due to absorption.

**Limitations:**

There are no concerning limitations or potential negative societal impacts from this work.

**Quality:**

3

**Strengths And Weaknesses:**

A note: as the saying goes, all models are wrong, but some are useful. I will pay most attention to whether or not feature absorption is a useful model for explaining how SAEs work, and how that in turn affects the usefulness of SAEs as a model to explain how LLMs work.

Strengths
1. A substantial amount of evidence is shown to convincingly support that feature absorption occurs in SAEs used in practice.
2. The feature absorption can be replicated in small toy examples to be studied.
3. Feature absorption can be theoretically shown to be preferred by the SAE training procedure.
4. If feature absorption indeed occurs, it is probably a widespread phenomenon across many open-source SAEs, significantly hindering mechanistic interpretatability efforts.

Weaknesses
1. It is unclear to me that describing a specific kind of feature splitting as "feature absorption" helps to explain more about what is happening to the SAE than just saying that one of the feature splits is extremely small. I am unsure that the identification of feature absorption as a separate phenomenon from feature splitting adds that much interpretation value when the interpreter already understands that feature splitting is common and can make the SAE explain things less clearly sometimes. If we could just redefine one of the "true human-interpretable features" to be the union of the split features, this would produce hierarchical co-occurrence of features leading to feature absorption which wasn't there in the first place. Individual feature absorptions may only occur under certain allotments of "true features" and not others, making the presence/absence of this problem inherently subjective.
2. In order to evaluate the test metric for feature absorption, you need to have a probe, and that probe must be trained with labeled data corresponding to the feature you are testing for. You could only do this probing successfully if you already knew the feature you are looking for is there in the model. But then, why would you even be using the SAE and feature absorption metric in the first place? You've already found the feature's direction using your probe, which I thought was the whole point of doing all of that. Alternatively, maybe you don't know the features and you're using the SAE to find them. Then, you can't be training a probe to make your absorption metric, since you don't know what the feature is that you want to label your data for. Again, the metric seems to be useless. It seems like the test metric can play no actual role in any part of the core pipeline of model interpretation.

---

> ### Author Rebuttal · Authors · 2025-07-30
>
> We thank the reviewer for their engagement with our work and thoughtful questions. We will address the minor issues raised in the final version of the paper. We address the main concerns below:
>
> > It is unclear to me that describing a specific kind of feature splitting as "feature absorption" helps to explain more about what is happening to the SAE than just saying that one of the feature splits is extremely small … If we could just redefine one of the "true human-interpretable features" to be the union of the split features …
>
> This is an important point to clarify to understand the significance of feature absorption compared with the traditional understanding of feature splitting, and why absorption is so pernicious to interpretability efforts. The traditional understanding of feature splitting is that in a small SAE, a more coarse-grained feature like “country” will be split into more specific features for every country like “France”, “Japan”, etc in a larger SAE. If this were all that happens in feature splitting, this is not a problem for interpretability as all features in both cases are perfectly interpretable and understandable.
>
> However, what the feature absorption work shows is that when there are hierarchical features like “country” and “Japan” (since Japan is a country), we instead end up with a “country” latent that fires on most countries, but, mysteriously, not on some very specific countries like “Japan”. This harms the interpretability of the “country” latent, since it is not firing in places where it seemingly should fire. Can we even claim that this is a “Country” latent at this point? Even worse, since we do not have ground-truth data for the “true features” of a model, we have no way to know when or if this is happening. We cannot trust that any latent actually captures the concept it seems like it captures. For instance, even our “Japan” latent may fail to fire on mentions of “Osaka” if “Osaka” has its own latent in the SAE. Even worse, imagine we think we find a latent for “the model is being dishonest” and use it to monitor the model, but due to feature absorption, this latent fails to fire on specific types of dishonesty so our monitor silently fails.
>
> If we knew which latents in the SAE were engaging in splitting and absorption, we could indeed create grouped “super-latents” as the union of all split latents, as you suggest, but in real LLMs we do not have ground-truth feature data, and do not yet have a way to automatically detect all feature splits in a given SAE. Indeed, we know of researchers manually trying to cluster SAE latents that seem related as an interim attempt to do this, but automatically detecting/grouping of feature splitting is still an unsolved research problem.
>
> > My Weakness item #2 comes from my understanding that the purpose of the absorption metric is that it should be used in conjunction with the SAEs when trying to identify the unknown features that may be present in a model. Is this the intended purpose or use case? It is unclear whether the metric is supposed to serve a different purpose instead, e.g., to help with further interpretation of the model after the features have already been identified. If you would explain an example of where you expect this metric to be useful in a different part of the interpretation pipeline, I may increase my score.
>
> Our absorption metric is not meant to be used to identify every specific instance of absorption in an SAE, as like you rightly point out, that would require already having a probe for all the features captured by the SAE and defeats the purpose of using an SAE. Instead, our absorption metric is meant to do the following two things:
>
> **Prove that absorption happens in SAEs trained on real LLMs.**
>
> The goal of this paper is to introduce the problem of feature absorption in SAEs. We show the theoretical underpinnings of absorption in simple toy models, but we need a way to demonstrate whether this happens or not in SAEs trained on real LLMs. Our metric fills that purpose, showing that all LLM SAEs we tested engage in feature absorption. This proves that absorption is not only a theoretical problem but can be found in real SAEs that are used by researchers as well.
>
> **Guide the search for new SAE architectures that do not engage in feature absorption.**
>
> The best outcome of this work would be to inspire new SAE architectures that do not engage in feature absorption. However, this requires that we have a way to benchmark the absorption rate of a given SAE, so we can tell if a new SAE architecture no longer engages in absorption. Notably, this does not require that every instance of absorption across every concept / latent in the SAE be identified by our metric, but only that we can gauge the relative rate of absorption across a few concepts to use as a benchmark.
>
> In this role as a benchmark, our absorption metric is similar to any other benchmark used throughout the ML field. For instance, a benchmark like HaluEval [1] is meant to benchmark hallucinations in LLMs. HaluEval cannot tell you if an arbitrary LLM output is a hallucination or not, but an LLM that does not hallucinate should score well on the HaluEval benchmark. Similarly, our absorption metric cannot tell you if any given SAE latent contains absorption or not, but we expect an SAE that does not engage in absorption to score well on our absorption metric.
>
> ### References
>
> [1] Li, Junyi, et al. "HaluEval: A Large-Scale Hallucination Evaluation Benchmark for Large Language Models." Proceedings of the 2023 Conference on Empirical Methods in Natural Language Processing. 2023.

---

> ### Comment · Reviewer_64Na · 2025-08-06
>
> **On the first point Weakness #1:** I can see the point that when the LLM truly has hierarchical features, then it would be bad for an SAE to absorb some of an otherwise well-formed latent into some special-case latents. My concern is rather what happens in a scenario where the LLM's true internal representation is really actually one feature for "countries that are not Japan" and one feature for "Japan". It would be hard for us to believe that this is the way it indeed chooses to process the data, but nonetheless if it happens, surely we would want to know. If this were to happen, and an SAE pointed out "countries that are not Japan" and "Japan" as separate latents, as is the truth in this scenario, then we would be turning a blind eye to dismiss this observation as feature absorption and think that obviously there should really be a "countries" latent instead. This would be especially painful because in such a scenario, the true explanation seems otherwise very difficult to find, the SAE is serving it to us on a platter, and we are using the idea of absorption to just ignore what the SAE says about how the model works and call it wrong.
>
> I have just given a straw man argument for why we might want the SAE to behave as originally, with the "feature absorption" included. But notice that the ground truths for the two possible situations, one where the LLM uses a feature for "countries that are not Japan" and one for "Japan", and the other where the LLM uses a feature for "countries" and one for "Japan", are equivalent up to a linear transformation. It's not really true that one explanation of what the LLM is doing, as derived by the SAE, is fundamentally "more right" or even "different" than the other; this distinction only arises once the human who does the interpreting asserts that "countries" is a realistic thing that the LLM might have learned a feature for while "countries that are not Japan" is not. I don't think we should be as biased as we are towards one explanation over the other. In this way, I argue that it is somewhat subjective whether or not absorptions in SAEs are problematic, unless we have some more evidence of otherwise.
>
> **On the second point Weakness #2:** Got it, that makes sense. My concern has been resolved and I will increase my score by 1.

---

> > ### Author Response · Authors · 2025-08-06
> >
> > We thank the reviewer for their continued engagement and for raising their score. We address their further concern below:
> >
> > > I can see the point that when the LLM truly has hierarchical features, then it would be bad for an SAE to absorb some of an otherwise well-formed latent into some special-case latents. My concern is rather what happens in a scenario where the LLM's true internal representation is really actually one feature for "countries that are not Japan" and one feature for "Japan". It would be hard for us to believe that this is the way it indeed chooses to process the data, but nonetheless if it happens, surely we would want to know.
> >
> > We agree that the goal of using SAEs is to discover the true features expressed by the LLM, regardless of whether those features are intuitive to humans or not. This highlights the urgency of finding new SAE architectures that do not engage in feature absorption, so that when the SAE has a latent that seems to track "countries that are not Japan" and another latent for "Japan", we can have confidence that this is just how the LLM represents concepts rather than being an artifact of feature absorption.
> >
> > > It's not really true that one explanation of what the LLM is doing, as derived by the SAE, is fundamentally "more right" or even "different" than the other; this distinction only arises once the human who does the interpreting asserts that "countries" is a realistic thing that the LLM might have learned a feature for while "countries that are not Japan" is not. I don't think we should be as biased as we are towards one explanation over the other. In this way, I argue that it is somewhat subjective whether or not absorptions in SAEs are problematic, unless we have some more evidence of otherwise.
> >
> > Our toy model analysis in this paper (section 3) shows that current SAE architectures will engage in feature absorption any time there are hierarchical features represented in the underlying data the SAE is trained on. This analysis does not depend on any interpretation of the meaning of the features, and arises simply as an undesirable result of the sparsity penalty used to train SAEs. Thus, with current SAE architectures, we will struggle to differentiate between the case of latents "counties except Japan" and "Japan" arising due to feature absorption and arising due to it being just how the LLM works internally. This means we cannot trust the concepts learned by current SAEs reflect the true internal representations of the model, and this is a big problem for us being able to rely on SAEs to interpret LLMs.
> >
> > Our goal with this work is to make the community aware that current SAEs have this failure mode, and to motivate the search for new SAE architectures or training methods that do not suffer from feature absorption. If we can solve feature absorption, then we can get closer to the goal of uncovering the true representations used in LLMs, regardless of whether or not the LLM representations are intuitive to humans.

---

> > > ### Comment · Reviewer_64Na · 2025-08-08
> > >
> > > Thank you for your thoughtful response, I appreciate your clarifications. I will keep my score of 4.

---

### Official Review · Reviewer_mChz · 2025-06-27

**Clarity:** 3
**Significance:** 3
**Originality:** 3
**Rating:** 5
**Confidence:** 4

**Summary:**

This paper introduces and systematically analyzes a phenomenon that can arise in sparse autoencoders (SAEs) when interpreting the activation space of large language models (LLMs), termed feature absorption by the authors. When hierarchical features exist in the model (e.g., “starts with S” and “short”), the SAE’s sparsity-driven optimization may cause the representation of higher-level features to be absorbed into more specific latents, resulting in the general latent failing to activate where it intuitively should. The authors propose a metric—absorption rate—to quantify this effect, and empirically demonstrate its prevalence across various models and layers. This work offers a new perspective and raises important questions about the reliability and applicability of SAEs in interpretability tasks.

**Questions:**

- Any thoughts on when feature absorption is more likely to occur, and whether there are ways to mitigate it? For example, are certain types of features or training settings more susceptible to this effect?

- When multiple features with overlapping semantics co-occur, stronger latents might dominate weaker ones even without true absorption. Clarifying how such dominance is distinguished from actual absorption would strengthen the analysis.

- In my opinion, SAEs may serve as a form of knowledge compression, which can introduce ambiguity. While the resulting features are more interpretable than individual neurons, activation-based features descriptions might still underrepresent a feature’s full role—especially in cases where features reflect different functional categories, such as input features, which capture patterns within the model’s input, and output features, whose main role is to directly influence the tokens the model generates.

**Ethical Concerns:**

["NO or VERY MINOR ethics concerns only"]

**Final Justification:**

Thank you for the detailed responses. I find this work particularly valuable for deepening our understanding of the features extracted from Sparse Autoencoders (SAEs). The insights provided contribute meaningfully to the field, and I am genuinely interested in the potential of this approach.

**Limitations:**

yes

**Paper Formatting Concerns:**

None.

**Quality:**

4

**Strengths And Weaknesses:**

Strengths:

- The paper introduces a novel and clearly defined phenomenon—feature absorption—that highlights how sparsity objectives in SAEs can interfere with the reliable representation of general features in LLMs.

- The empirical study is well-executed across multiple models and layers, including Gemma, Qwen2, and LLaMA variants, and provides consistent evidence for the prevalence of absorption effects.

- The proposed absorption rate offers a simple and interpretable metric to quantify this behavior, and the combination of toy examples and real-model evaluations helps ground the intuition.

- The work raises important questions for the future design of interpretable sparse representations and may inform efforts toward more robust circuit discovery.

Weaknesses:

- The experiments focus on relatively small-scale models; it remains unclear whether similar absorption patterns persist in larger models such as 7B or 9B.

- The absorption rate metric relies on token-level ablation and is mainly applicable to lexical features, limiting its ability to capture absorption in higher-level or cross-token concepts.

---

> ### Author Rebuttal · Authors · 2025-07-30
>
> We thank the reviewer for their thoughtful reading and review of the paper. We address the questions below:
>
> > Any thoughts on when feature absorption is more likely to occur, and whether there are ways to mitigate it? For example, are certain types of features or training settings more susceptible to this effect?
>
> Qualitatively, absorption seems most prevalent for features that are very “general”. For example, we have never found clear “parts of speech” latents in any SAE despite LLMs clearly representing this information, likely because latent tracking parts of speech are too riddled with absorption exceptions to even be recognizable anymore. We find that latents for concepts for anything that’s a “category” or “class of thing” also seems to be very susceptible to absorption in our experience, e.g. things like “is Python code” or “is an animal”, etc...
>
> So far, we have found that just being aware of absorption has been very helpful when using SAEs for interpretability work. For example, we no longer assume that an SAE latent for a concept not firing when we think it should therefore means the concept is not present in the LLM activations. We also no longer assume that our inability to find a clear SAE latent for a given concept is therefore evidence that the model does not represent that concept - it could also be that absorption has simply made the latent for the concept not fire enough that we can no longer tell what that latent mainly is about. We also try to use supervised methods like probes whenever feasible as these do not suffer from absorption. Despite this, we do still find SAEs very useful as a first-pass for interpretability work, as they still often surface concepts that are unexpected and can be further targeted with supervised probes.
>
> > In my opinion, SAEs may serve as a form of knowledge compression, which can introduce ambiguity. While the resulting features are more interpretable than individual neurons, activation-based features descriptions might still underrepresent a feature’s full role—especially in cases where features reflect different functional categories, such as input features, which capture patterns within the model’s input, and output features, whose main role is to directly influence the tokens the model generates.
>
> We have also seen recent work that dives into this dichotomy of input vs output features in SAEs [1]. It could be that separating classification features and generation features is core to how LLMs represent concepts internally. Regardless, it certainly seems to be a promising direction of future work to try to understand the types of features that SAEs learn, and if there are further relationships or geometry within features captured by SAEs.
>
> ### References
>
> [1] Arad, Dana, Aaron Mueller, and Yonatan Belinkov. "SAEs Are Good for Steering--If You Select the Right Features." arXiv preprint arXiv:2505.20063 (2025).

---

> ### Comment · Reviewer_mChz · 2025-08-05
>
> Thank you for the responses. While I appreciate the clarifications, I will maintain my original score for now.

---

### Official Review · Reviewer_Uf4h · 2025-07-03

**Clarity:** 4
**Significance:** 4
**Originality:** 4
**Rating:** 5
**Confidence:** 4

**Summary:**

The paper demonstrates theoretically and empirically a potential mismatch between what we would ideally "want" from an SAE decomposition, versus what the objective function of the SAE prefers in reality.

Specifically, the authors conjecture the existence of the *feature absorption* phenomenon. Suppose there are two features in the data, $G$ (a more "general" one) and $S$ (a more "specific" one), such that whenever $S$ is present in an input, so is $G$. Let $f_S, f_G$ be the "true" feature vectors corresponding to $S, G$ in activation space. The main observation behind absorption is that, in this setup, the SAE objective will prefer to model this situation using two latents $v_G = f_G, v_S= f_S + f_G$ such that, when $G$ is present by itself, $v_G$ will be active in the SAE, and when $S$ (and therefore $G$) is present, *only* $v_S$ will be active. This solution is sparser than allocating a latent for each of $f_G,f_S$ and firing both when $S$ is present.

The authors show this plays out in a simple toy model, and in a real-world setup where basically $G=$ "token starts with some letter" and $S=$ "token equals a specific token that starts with that letter". They define a formal measure of absorption and investigate how different SAE parameters and metrics influence its prevalence according to that measure.

**Questions:**

- Def. of hierarchical features (line 54) is unclear. What does it mean for a feature to "fire"? I think there are some implicit assumptions being made here, and it would benefit the paper to make them more explicit.
- is it really necessary to train the toy model on 100M activations (Section 3, line 87)?
- plot 6b would perhaps be more interpretable if we sort the x-axis by the y-axis value?

Typos and other minor nitpicks:
- fig 1 right: "starts S" should be "starts with S"
- line 87: should probably use L1 or $\ell_1$ for both occurrences instead of L1/l1
- line 103: "This is exactly the sort of gerrymandered feature firing pattern we saw in real SAEs for the starting letter task" - we have not seen this yet in the paper! Maybe say "we will see later in Section X"
- line 178: "We conduct an ablation experiment on the `_short` token" - might help the reader to remind them of what an "ablation experiment" means in the context of this paper, or to altogether give this intervention a more precise name. "ablation experiment" in the current paper is used to point to a very specific technical term; but it's also a general term in the ML literature, which may give rise to confusion.

**Ethical Concerns:**

["NO or VERY MINOR ethics concerns only"]

**Final Justification:**

I maintain my recommendation for acceptance after the rebuttal period.

**Limitations:**

As pointed out, it would be interesting to demonstrate absorption in other settings with more practically interesting hierarchical features.

**Quality:**

4

**Strengths And Weaknesses:**

This paper does a great job at illuminating a non-obvious qualitative phenomenon in interpretability, and explaining and demonstrating it in a quite intuitive and reasonable progression of thought experiments, toy models and real-world experiments. Despite the somewhat constrained real-world setup being studied (first-letter features are perhaps not the ultimate reason we as a field want to develop SAEs), the authors argue very persuasively for the general plausibility of feature absorption and its implications on the use of SAEs for interpretability. The paper is at times rough around the edges, but I'm certain this can be largely overcome by a small revision.

Overall, this is a timely and significant contribution to the SAE line of work.

---

> ### Author Rebuttal · Authors · 2025-07-30
>
> We thank the reviewer for their thorough reading and understanding of the paper, and for the feedback and suggestions. We will fix all typos in the camera-ready version of the paper - thank you for noting these. We address the questions below:
>
> > Def. of hierarchical features (line 54) is unclear. What does it mean for a feature to "fire"? I think there are some implicit assumptions being made here, and it would benefit the paper to make them more explicit.
>
> By “fire” we just mean the feature has a non-zero activation value. Our model is that a feature is either firing, meaning it has a non-zero value, or is not firing and thus has zero activation value. We will clarify this further in the text.
>
> > is it really necessary to train the toy model on 100M activations (Section 3, line 87)?
>
> It’s almost certainly overkill to train on this many activations for this toy setup, but since the toy model and SAE are both so small, it only takes a few minutes for this to run on CPU.
>
> > plot 6b would perhaps be more interpretable if we sort the x-axis by the y-axis value?
>
> Thank you for this feedback. Our goal here was just to convey the rough ranges of cos similarities of SAE latents with the probe, and where the core latents we are interested in fit in. We will experiment with different layouts for this plot for the camera-ready version, as we agree it can be improved.

---

> > ### Comment · Reviewer_Uf4h · 2025-08-07
> >
> > Thank you for your responses to my questions

---

### Official Review · Reviewer_Bisd · 2025-07-05

**Clarity:** 3
**Significance:** 3
**Originality:** 4
**Rating:** 5
**Confidence:** 4

**Summary:**

Sparse autoencoders (SAEs) are an interpretability tool that aim to decompose language model activations into a linear combination of individually-interpretable features. SAEs allow for a large set of possible features (the “dictionary”) but require that any individual activation may only use a small number of these features (a sparsity constraint), and hypothesize that this will cause the learned features to be interpretable.

This paper presents an obstacle to the interpretability property: when in reality there are hierarchical features, the optimization objective for SAEs encourages “feature absorption”: the coarser feature learns not to fire when the more fine-grained feature fires, so as to improve sparsity. Specifically, suppose there are two features A and B where if B fires then A should fire. Then instead of learning [A] and [B] as separate features, the SAE can learn [A and not B] and [B] as separate features. Then on inputs where both A and B are present, the SAE only fires one feature instead of two, doing better on sparsity.

The paper demonstrates this in a toy setting where the ground truth can be controlled, and then studies it in practice on the “first letter task” (“dog” → “d” and so on). For this task, there are often a small number of SAE latents capturing the first letter information, but that then fail to fire on some common tokens where the interpretation would suggest they should fire. For example, the “starts with s” feature fails to fire on the “short” token, even though a linear probe can perform the “starts with s” task well. Ablation experiments demonstrate that this information is instead in a feature that fires on “short” only, suggesting that the “short” feature has absorbed the “starts with s” feature, in accordance with theory.

The authors define a feature absorption metric in this task, and find that lots of SAEs have significant feature absorption on this task, with rates particularly high for wider and sparser SAEs.

**Questions:**

How much do you expect feature absorption to be a problem in practice for typical applications of interpretability? Can you quantify this in some way?

**Ethical Concerns:**

["NO or VERY MINOR ethics concerns only"]

**Final Justification:**

I stand by the summary from my original review:

> Overall, this is a great paper that changed my views about how promising SAEs are. Feature absorption is an obvious problem with SAEs once pointed out, but I hadn’t thought about it before. I wish there was more investigation and evidence about how large a problem is – for example, perhaps one could brainstorm ten different pairs of hierarchical features, and manually check for how many of them feature absorption has happened. But even without this quantification it’s a valuable paper that should be accepted.

**Limitations:**

Mostly yes, though I think they should also acknowledge the limitation of focusing just on the first-letter task.

**Quality:**

3

**Strengths And Weaknesses:**

## Strengths

1. The core problem with SAEs identified in this paper is very intuitive and seems like it should apply quite widely – yet to my knowledge it was not known in the literature. (It may have been known in the dictionary learning or sparse coding communities; I am not familiar with that literature.)
2. The authors have a good blend of toy models and concrete examples that build intuition and demonstrate the mechanism of feature absorption, along with a larger scale quantitative study that quantifies how large a problem this is in practice.

## Weaknesses

1. As the authors acknowledge, the feature absorption metric is likely unreliable in a variety of ways. For example, it relies on ablation on the original token, which will often be ineffective in later model layers since the information will already have been moved by an attention head to the final token position. In addition the metric has multiple hyperparameters that are judgment calls; changing these would presumably affect the results a decent amount, and there isn’t a particular justification for the choices made.
2. The empirical results are almost entirely based on the first-letter task, which makes it hard to know how large the feature absorption problem will be in general. It is possible that the more abstract features that are more often the goal of interpretability research would not have the same issue, though the theoretical / conceptual argument remains strong so I consider this to be a fairly minor weakness.
3. The paper identifies a problem with SAEs but doesn’t propose any methods for solving that problem, limiting its impact. (However, SAEs are popular enough that pointing out a significant problem with SAEs is still a very valuable contribution, even if it doesn’t come with a solution.)

## Summary

Overall, this is a great paper that changed my views about how promising SAEs are. Feature absorption is an obvious problem with SAEs once pointed out, but I hadn’t thought about it before. I wish there was more investigation and evidence about how large a problem is – for example, perhaps one could brainstorm ten different pairs of hierarchical features, and manually check for how many of them feature absorption has happened. But even without this quantification it’s a valuable paper that should be accepted.

---

> ### Author Rebuttal · Authors · 2025-07-30
>
> We thank the reviewer for their thorough reading of the paper and for summarizing the core ideas so well. We will add the focus on first-letter to the limitations section for the CR version of the paper. We address the question below:
>
> > ​​How much do you expect feature absorption to be a problem in practice for typical applications of interpretability? Can you quantify this in some way?
>
> While it is difficult to quantify exactly how big of a problem absorption is, qualitatively we have found it to be a significant barrier to using SAEs for most interpretability tasks we have tried to use them for. We list a few tasks we have personally encountered, and how absorption has been problematic below:
>
> ### SAE latent ablation
> One common task in interpretability is to detect how much a LLM relies on a given concept when performing a task. For instance, we may want to detect how much the LLM is relying on gender bias when performing a classification task. A natural way to use a SAE for this is to find a latent representing “male” or “female” and force its activation to 0 when running the model, and then record how this changes model outputs. However, in practice we find this almost never works, we suspect due to feature absorption. Likely the “gender” direction is encoded into more specific latents, e.g. for a specific name like “Lucy” or “George” mentioned in the prompt, and thus ablating the more general “gender” SAE latent has little or no effect on the model output, even if the model is using the underlying concept.
>
> ### Feature attribution
> Another common task is to use feature attribution to understand which concepts a model is using internally in order to perform a task. For instance, we may want to know how a LLM knows that “London” is located in “England”. We can use an SAE for this, inserting the SAE into the forward pass of the model and performing backprop on the “ England” token logit to see which SAE latents receive a large attribution score. However, due to feature absorption, this often does not locate the concepts we expect it should. For instance, we may find the SAE has a “London” feature that fires strongly and gets all attribution, and hides the more general parent features like “located in England”, even though the SAE has a latent which appears to be a “located in England” latent. The result is that it becomes hard to tell with certainty what concepts the model is relying on.
>
> In our experience, we have found knowledge of feature absorption to be essential for any task where we use an SAE. We still find SAEs valuable as a first-pass technique as they still will often surface things we did not expect, but we use them with the knowledge that absorption means often things that should fire may not fire, and to not be surprised by this and conclude the model is not using some concept as a result. We also choose to rely on supervised methods like probes over SAEs when possible, as supervised probes do not suffer from absorption. We have also found it helpful to manually group latents that seem semantically similar together for interpretability tasks, again relying on the knowledge that a latent we expect to fire may not fire, but that we can sometimes get around this by manual grouping.

---

### Comment · Area_Chair_RHNp · 2025-08-06

Dear Reviewers,
After reading the authors' responses, have your concerns been addressed? Do you have follow-up questions? Do you want to adjust your ratings? If you have not, please remember to acknowledge that you have read the authors' response.
Best,
AC

---

### Decision · Program_Chairs · 2025-09-17

**Decision:**

Accept (oral)

**Comment:**

Summary
This paper studies the feature splitting in SAE: as the number of features increase in SAE, hierarchical features tend to split into finer features. However, this paper shows that this feature splitting is not robust. Seemingly monosemantic features fail to fire where they should, and instead get absorbed into their children features. This paper introduces a metric to detect feature absorption, and validate the findings on hundreds of LLM SAEs. This paper shows theoretically why sparse dictionary learning encourages feature absorption. This paper argues that varying SAE sizes or sparsity is insufficient to solve this issue, and discuss potential solutions.

Strengths
- The problem of feature absorption in feature splitting is intuitive and, despite should apply quite widely, was not present in the literature.
- This paper uses a combination of toy models and concrete examples and larger-scale studies to demonstrate the mechanism of feature absorption and how large this problem is.
- The feature absorption can be theoretically shown to be preferred by the SAE training procedure (e.g., in Appendix A.2).
- From Bisd: This feature absorption problem, once identified, can change people's views about how promising SAEs are.
- From Uf4h: The authors argue very persuasively for the general plausibility of feature absorption and its implications on the use of SAEs for intepretability.
- From mChz: This work is particularly valuable for deepening our understanding of the features extracted from SAEs.

Weaknesses
- The feature absorption metric and the empirical results focus on the first-letter task, and the generalizability to other tasks would be good to have.
- Minor polish could be made to improve the presentation of this paper.